# Leaf nitrogen from first principles: field evidence for adaptive variation with climate

**Ning Dong[1,3], Iain Colin Prentice[1,2], Bradley J. Evans[1,3,4], Stefan Caddy-Retalic[5,6], Andrew J. Lowe[5,6,7], Ian J. Wright[1]**

[1]Department of Biological Sciences, Macquarie University, North Ryde, NSW 2109, Australia

[2]AXA Chair of Biosphere and Climate Impacts, Department of Life Sciences, Imperial College London, Silwood Park Campus, Buckhurst Road, Ascot SL5 7PY, UK

[3]Terrestrial Ecosystem Research Network: Ecosystem Modelling and Scaling Infrastructure, University of Sydney, NSW 2006, Australia

[4]Faculty of Agriculture and Environment, Department of Environmental Sciences, University of Sydney, NSW 2006, Australia

[5]Terrestrial Ecosystem Research Network: Australian Transect Network, University of Adelaide, North Terrace, Adelaide, SA 5005, Australia

[6]School of Biological Sciences and Environment Institute, University of Adelaide, North Terrace, Adelaide, SA 5005, Australia

[7]Science, Monitoring and Knowledge Branch, Department of Environment, Water and Natural Resources, Hackney Road, Kent Town, SA 5005, Australia

*Correspondence to*: Ning Dong (ning.dong@students.mq.edu.au)

**Abstract**

Nitrogen content per unit leaf area ($N_{area}$) is a key variable in plant functional ecology and biogeochemistry. $N_{area}$ comprises a structural component, which scales with leaf mass per area (LMA), and a metabolic component, which scales with Rubisco capacity. The co-ordination hypothesis, as implemented in LPJ and related global vegetation models, predicts that Rubisco capacity should be directly proportional to irradiance but should decrease with increases in $c_i$:$c_a$ and temperature because the amount of Rubisco required to achieve a given assimilation rate declines with increases in both. We tested these predictions using LMA, leaf $\delta^{13}$C and leaf N measurements on complete species assemblages sampled at sites on a North-South transect from tropical to temperate Australia. Partial effects of mean canopy irradiance, mean annual temperature and $c_i$:$c_a$ (from $\delta^{13}$C) on $N_{area}$ were all significant and their directions and magnitudes were in line with predictions. Over 80% of the variance in community-mean (ln) $N_{area}$ was accounted for by these predictors plus LMA. Moreover, $N_{area}$ could be decomposed into two components, one proportional to LMA (slightly steeper in N-fixers), the other to Rubisco capacity as predicted by the co-ordination hypothesis. Trait gradient analysis revealed $c_i$:$c_a$ to be perfectly plastic, while species turnover contributed about half the variation in LMA and $N_{area}$.

Interest has surged in methods to predict continuous leaf-trait variation from environmental factors, in order to improve ecosystem models. Coupled carbon-nitrogen models require a method to predict $N_{area}$ that is more realistic than the widespread assumptions that $N_{area}$ is proportional to photosynthetic capacity, and/or that $N_{area}$ (and photosynthetic capacity) are determined by N supply from the soil. Our results indicate that $N_{area}$ has a useful degree of predictability, from a *combination* of LMA and $c_i$:$c_a$ – themselves in part environmentally determined – with Rubisco activity, as predicted from local growing conditions. This finding is consistent with a 'plant-centred' approach to modelling, emphasizing the adaptive regulation of traits. Models that account for biodiversity will also need to partition community-level trait variation into components due to phenotypic plasticity and/or genotypic differentiation within species, versus progressive species replacement, along environmental gradients. Our analysis suggests that variation in $N_{area}$ is about evenly split between these two modes.

# 1 Introduction

Nitrogen (N) is an essential nutrient for primary production and plant growth, and nitrogen content per unit leaf area ($N_{area}$) is a key variable in plant functional ecology and biogeochemistry. A strong correlation between leaf N and photosynthetic capacity has been observed, and is to be expected because typically almost half of the N in leaves is invested in the photosynthetic apparatus (Field and Mooney 1986; Evans and Seemann 1989; Evans 1989). This component of $N_{area}$ is approximately proportional to the maximum rate of carboxylation ($V_{cmax}$) at standard temperature, also expressed per unit area (Wohlfahrt et al. 1999; Takashima et al. 2004; Kattge et al. 2009). Cell walls account for a further significant fraction of leaf N (Lamport and Northcote 1960; Niinemets and Tenhunen 1997; Onoda et al. 2004). Leaf mass per area (LMA) is positively correlated with cell-wall N (Onoda et al. 2004) and is used as an index of plant investment in cell-wall biomass (Reich et al. 1991; Wright and Cannon 2001). Thus, $N_{area}$ can usefully be considered as the sum of a 'metabolic' component related to $V_{cmax}$ and a 'structural' component proportional to LMA. Leaves with high $V_{cmax}$ usually have high LMA and so these two quantities can be at least partially correlated, as seen clearly (for example) in parallel vertical gradients of $V_{cmax}$ and LMA within canopies of one species (e.g. Niinemets and Tenhunen 1997). Across different species and environments, however, there is scope for considerable independent variation in $V_{cmax}$ and LMA, implying the need to consider them separately.

Dynamic Global Vegetation Models (DGVMs) are being extended to include interactive carbon (C) and N cycles (Thornton et al. 2007; Xu-Ri and Prentice 2008; Zaehle and Friend 2010). But there remain many open questions about the implementation of C-N coupling (Prentice and Cowling 2013), including the control of leaf N content, which is treated quite differently by different models. For example, one common modelling approach predicts photosynthetic capacity from $N_{area}$, and $N_{area}$ in turn from soil inorganic N supply (e.g. Luo et al. 2004). This implies an assumption that the soil environment, and soil microbial activity in particular, are the primary controls of $N_{area}$ and photosynthetic capacity at the leaf level. An alternative assumption is that photosynthetic capacity is optimized as a function of irradiance, leaf-internal $CO_2$ concentration ($c_i$) and temperature (Haxeltine and Prentice 1996, Dewar 1996) – implicit in the widely used LPJ DGVM (Sitch et al. 2003) and other models derived from it, including LPJ-GUESS (Smith et al. 2001) and LPX (Prentice et al. 2011a; Stocker et al. 2013). This 'plant-centred' approach embodies the idea that plant allocation processes

(and thus, not soil microbial processes) determine leaf-level traits. Limited N supply, by this reasoning, should lead to the production of fewer leaves, rather than leaves with suboptimal capacity. More specifically it is derived from a long-standing concept, the 'co-ordination hypothesis', which states that the Rubisco- and electron transport-limited rates of photosynthesis tend to be co-limiting under average daytime conditions (Chen et al. 1993; Haxeltine and Prentice 1996; Maire et al. 2012). Co-limitation is optimal – even though mechanistically, it may be an inevitable outcome of leaf metabolism (Chen et al. 1993) – in the sense that it provides the right balance of investments in the biochemical machineries for carboxylation and electron transport. It implies that enzyme activities adjust, over relatively long periods (weeks or longer), so that co-limitation holds. An important consequence is that the predicted responses of photosynthetic traits and rates to environmental variables observed in the field (whether temporally, comparing different seasons or spatially, comparing different environments) are substantially different from those seen in short-term laboratory experiments. Specifically, $V_{cmax}$ (and thus the metabolic component of $N_{area}$) is predicted to be directly proportional to irradiance; to decrease with increasing $c_i$:$c_a$; and to decrease with increasing temperature. These predictions are supported in general terms by an observed positive relationship between $N_{area}$ and irradiance (Field 1983; Wright et al. 2005), a negative relationship between $N_{area}$ and $c_i$:$c_a$ (Wright et al. 2003; Prentice et al. 2011b; Prentice et al. 2014), and (in woody evergreens at least) a negative relationship between $N_{area}$ and temperature (845 species: data from Wright et al. 2004). But there has been no systematic attempt to quantitatively assess the relationship of leaf N to environmental and structural predictors across environmental gradients. Such empirical work is needed to assess and underpin methods of C-N cycle coupling in DGVMs.

Here we set out to test the predictability of $N_{area}$ using measurements carried out on dried plant material collected by the Terrestrial Ecosystem Research Network (TERN) AusPlots and Australian Transect Network facilities, at 27 sites on a north-south transect across the Australian continent. The transect extended from the wet-dry (monsoonal) tropics to the dry-wet (mediterranean) temperate zone via the arid interior, and encompassed substantial variation in all of the hypothesized controls of $N_{area}$ (Fig. 1). The Ausplots protocol involves sampling all species within a 100×100 m plot (White *et al*. 2012). We measured $N_{area}$, $\delta^{13}$C and LMA on all species at each site, and tested and quantified the effects of irradiance, $c_i$:$c_a$ ratio (from $\delta^{13}$C), temperature, LMA, and N-fixation ability (26% of the species sampled were N-fixers), on variation in $N_{area}$. The sampling design also allowed us to implement the

trait gradient analysis method introduced by Ackerly and Cornwell (2007), which has been surprisingly little used to date. A growing body of field measurements shows extensive leaf-trait variation within species and PFTs (Kattge et al. 2011; Meng et al. 2015). Trait gradient analysis allows trait variation to be partitioned into a component due to variation within species and a component due to species replacement.

## 2 Materials and Methods

Our analyses are based on 442 leaf measurements representing all species found in a 100 m ×100 m plot at each of 27 sites on a broad North-South transect across Australia (Fig. 1) We performed a regression analysis to test the relationships of $N_{area}$ to mean annual temperature (MAT), irradiance, plant traits leaf mass per area (LMA), $c_i$:$c_a$ ratio and N-fixation capacity. We also fitted a statistical model in which $N_{area}$ was treated as the sum of a metabolic component proportional to predicted (optimal) photosynthetic capacity at standard temperature (based on temperature, irradiance and $c_i$:$c_a$ ratio) and a structural component proportional to LMA. Finally, we carried out a trait gradient analysis in order to quantify the contributions of environment versus species identity to variation in $N_{area}$, $c_i$:$c_a$ ratio and LMA.

### 2.1 Climate data and analysis

Climatological data for the 27 sites were obtained from the eMAST/ANUClimate dataset (www.emast.org.au), which extends from 1970 to 2012 with 1 km spatial resolution across the entire continent. Mean annual precipitation (MAP) over this period at the sampling sites ranged from 154 to 1726 mm and mean annual temperature (MAT) from 14.1˚ to 27.6˚C. The moisture index (MI = $P/E_q$, where $P$ is mean annual precipitation and $E_q$ is equilibrium evapotranspiration, calculated with the STASH program: Gallego-Sala et al. 2012) varied from 0.07 to 0.82. The mean incident flux of photosynthetically active radiation (PAR) during daylight hours, expressed as photosynthetic photon flux density (μmol m$^{-2}$ s$^{-1}$), was also calculated using STASH. This incident flux (at the top of the canopy) was averaged through the canopy using Beer's law, as follows. First leaf area index ($L$) was estimated from remotely sensed (MODIS NBAR-derived using MOD43A4: http://remote-sensing.nci.org.au/u39/public/html/modis/fractionalcover-clw) fractional cover of

photosynthetic vegetation ($f_v$) in 1 km resolution at each site, from data assembled by the TERN AusCover facility (Guerschman et al. 2009):

$$L \approx -(1/k) \ln (1 - f_v) \tag{1}$$

where $k = 0.5$. Then absorbed PAR per unit leaf area ($I_L$) was calculated as:

$$5 \quad I_L \approx I_0 (1 - e^{-kL})/L \approx I_0 k f_v / \ln [1/(1 - f_v)] \tag{2}$$

where $I_0$ is the incident PAR above the canopy. This calculation yields $I_L \approx I_0$ for sparse vegetation ($L <$ 1) but $I_L$ becomes progressively smaller than $I_0$ as foliage density increases, reflecting the fact that the irradiance experienced by the average species is much lower in, say, a closed woodland than in an open shrubland, even if the PAR incident at the top of canopy is the same. In dense vegetation $I_L$ will underestimate the PAR exposure of canopy dominants and overestimate the PAR exposure of understory species. However, the use of a canopy average in this way was a necessary approximation (because we did not have quantitative information about the canopy position of each species) and considered preferable to using $I_0$, which will systematically overestimate PAR exposure for most species in a dense community.

## 2.2 Foliage sampling and analysis

Mature outer-canopy leaves of each species were sampled during the growing season using the AusPlots methodology (White *et al*. 2012). (Note that in denser vegetation many species sampled are in the understorey, so their 'outer-canopy' leaves are still shaded by the overstorey. Many species thus receive considerably reduced sunlight compared to the overstorey, implying that the canopy-average irradiance $I_L$ is more suitable than the top-of-canopy value $I_0$ as a community measure of irradiance.) In total, the 27 selected sites included 442 unique species, of which 37 were $C_4$ plants (not analysed further here). LMA was measured on the archived leaf samples by scanning and weighing the leaves. Subsamples (a mixture of material from at least 2 replicates) were analysed for C and N contents and bulk $\delta^{13}$C at the Stable Isotope Core Laboratory of Washington State University, USA. $N_{area}$ was calculated from N content and LMA. Carbon isotope discrimination ($\Delta$) values were derived from the reported $\delta^{13}$C values using the standard formula:

$$\Delta = (\delta_{air} - \delta_{plant})/(1 + \delta_{plant}) \tag{3}$$

where $\delta_{air}$ is the carbon isotope composition of air and $\delta_{plant}$ is the carbon isotope composition of the plant material. Because of the different diffusion rates and biochemical rates of carboxylation between $^{13}CO_2$ and $^{12}CO_2$, $\Delta$ can be used to estimate the $c_i:c_a$ ratio as:

$$c_i:c_a \approx (a + \Delta)/(b - a) \tag{4}$$

where the recommended standard values are $a = 4.4$ ‰ and $b = 27$ ‰ (e.g. Cernusak et al. 2013).

## 2.3 Analysis of $V_{cmax}$

Values of $V_{cmax}$ were predicted based on the co-ordination hypothesis, by equating the carboxylation- and electron transport-limited rates of photosynthesis and, as a simplifying assumption, treating the electron transport-limited rate as proportional to absorbed PAR (i.e. ignoring the saturation of the electron transport rate at high irradiances). These assumptions lead to the following estimate:

$$V_{cmax} \approx \varphi_0 I_L (c_i + K)/(c_i + 2\Gamma^*) \tag{5}$$

where $\varphi_0$ is the intrinsic quantum efficiency of photosynthesis (0.093: Long et al. 1993), $c_i$ is the leaf-internal concentration of $CO_2$, $K$ is the effective Michaelis-Menten coefficient of Rubisco, and $\Gamma^*$ is the photorespiratory compensation point. Values of both these quantities and their activation energies (governing their temperature responses) are based on the empirical *in vivo* determinations by Bernacchi et al. (2001), widely used in photosynthesis research. Both $K$ and $\Gamma^*$ were evaluated at standard atmospheric pressure and oxygen concentration, and site MAT. Predicted values of $V_{cmax}$ were adjusted to 25˚C, because the amount of N allocated to Rubisco and other enzymes involved in carboxylation should be proportional to $V_{cmax}$ at a standard temperature, not at the growth temperature.

## 2.4 Statistical methods

All statistics were performed in R3.1.3 (R Core Team 2015). Linear regressions were fitted using the *lm* function, partial residual plots were generated using the *visreg* package, and the relative contributions of different predictors were quantified using the Lindeman et al. (1980) method as implemented in the *relaimpo* package. In a first, exploratory statistical analysis, a linear model was fitted for ln $N_{area}$ with $c_i:c_a$, MAT, ln $I_L$, ln LMA and the factor 'N-fixer' as predictors. The regression slopes of ln $N_{area}$ against $c_i:c_a$, MAT and ln $I_L$ can all be independently predicted from the co-ordination hypothesis by

differentiation of eq (5) (see Appendix A. Note that these formulae explicitly predict the slopes for ln $N_{area}$). These predicted values were compared with the fitted values and their 95% confidence limits in order to assess support for the co-ordination hypothesis.

In a second analysis, community-mean values were calculated as simple averages across the species in each plot, omitting the factor 'N-fixer'. A linear model was fitted to the community means of ln $N_{area}$ as a function of $c_i{:}c_a$, MAT, ln $I_L$ and ln LMA to assess the predictability of leaf N at the community level.

In a third analysis, $N_{area}$ was modelled as a linear combination of the predictors Rubisco N, $N_{rubisco}$ (derived from predicted $V_{cmax}$ at 25˚C) and structural N, $N_{structure}$ (derived from LMA using the empirical relationship $N_{structure} = 10^{-2.67} \text{LMA}^{0.99}$, in g m$^{-2}$: Yusuke Onoda, personal communication 2015), including 'N-fixer' as a factor and allowing interactions of the predictors with this factor.

**2.5 Trait gradient analysis**

Trait gradients were generated for ln LMA, ln $N_{area}$ and $c_i{:}c_a$ following the analysis method of Ackerly and Cornwell (2007), again using simple averages across species to estimate community means. In this analysis species trait values were plotted against site-mean trait values. By definition, the regression of the species trait values against site-mean trait values has a slope of unity. For a perfectly plastic trait, regression of trait variation within species against the site-mean trait values would also yield a slope of unity. The common within-species slope that this approach provides is a measure of the fraction of trait variation due to phenotypic plasticity and/or genotypic variability. Its one-complement measures the fraction due to species turnover. Natural log transformation was applied to LMA and $N_{area}$ because of their large variance and skewed distributions, but not to $c_i{:}c_a$ because of its small variance and approximately normal distribution.

**3 Results**

**3.1 Leaf N variations with climate and leaf traits**

Significant partial relationships were found for ln $N_{area}$ *versus* $c_i{:}c_a$, MAT and ln $I_L$ (Table 1, Fig. 2). The relationship was negative for $c_i{:}c_a$, as expected because lower $c_i{:}c_a$ implies that a greater photosynthetic capacity is required to achieve a given assimilation rate (or equivalently: a stronger $CO_2$ drawdown is enabled by a higher $V_{cmax}$). The relationship was also negative for MAT, as expected

because there is an inverse relationship between temperature and the quantity of leaf proteins required to support a given value of $V_{cmax}$. The relationship was positive for ln $I_L$ (PAR), as expected because the higher the irradiance, the greater the carboxylation capacity required for co-limitation with the rate of electron transport.

Theoretical slopes for these relationships (derived in Appendix A) are compared with the fitted slopes in Table 1. For ln $N_{area}$ versus ln $I_L$, the theoretical slope is unity. The fitted slope of 0.874 (95% confidence limits: 0.685, 1.063) was statistically indistinguishable from unity. (A slope significantly greater than unity was found for ln $N_{area}$ versus ln $I_0$, i.e. top-of-canopy PAR, as expected as this measure underestimates the change in mean canopy PAR along the gradient from sparse, high-PAR to

dense, lower-PAR communities.) For ln $N_{area}$ *versus* $c_i:c_a$, the fitted slope of –0.611 (–1.107, –0.115) was fortuitously close to the theoretical slope of –0.615, although the value was only weakly constrained for these data. For ln $N_{area}$ *versus* MAT, the theoretical slope was obtained by subtracting the 'kinetic' slope of ln $V_{cmax}$ *versus* temperature (from the activation energy of carboxylation as given by Bernacchi et al. 2001) from the shallow positive slope implied by eq (5). The kinetic effect was

dominant, and results in an overall predicted negative slope of –0.048. The fitted slope of –0.047 (–0.060, –0.034) was indistinguishable from this theoretical slope, indicating acclimation to temperature by diminished allocation of N to metabolic functions at higher temperature, offsetting the increased reaction rate predicted by the Arrhenius equation. However this slope was shallower than would be predicted by the Arrhenius equation alone, reflecting the reduced quantum efficiency of

assimilation (a higher $V_{cmax}$ is required to support a given assimilation rate) at higher temperatures.

The proportion of leaf N allocated to Rubisco has generally been found to decline while the total N allocated to cell walls increases with increasing LMA (Hikosaka and Shigeno 2009). Fig. 2 shows a strong positive partial relationship between ln $N_{area}$ and LMA. N-fixers had generally higher $N_{area}$ than non-N-fixers (Fig. 2e: p < 0.001). The predictors together explained 55% of the variation in leaf N

across species and sites.

Fully 82% of the variation in the community-mean value of ln $N_{area}$ could be explained by the combination of community-mean LMA and environmental variables. Significant partial relationships of community-mean ln $N_{area}$ with MAT, ln $I_L$ and ln LMA (Table 2) were consistent with the results obtained at species level. The fitted slopes of ln $N_{area}$ against ln $I_L$ and MAT were again

indistinguishable from the theoretical values, albeit with wide error bounds due to the much smaller sample size (27 as opposed to 405). The community-level partial relationship between ln $N_{area}$ and $c_i{:}c_a$ showed a negative slope as predicted, although this relationship was barely significant ($p \approx 0.1$) due to the small sample size.

## 3.2 Leaf N as the sum of metabolic and structural components

Highly significant ($p < 0.001$) positive relationships were found between $N_{area}$ and the predicted Rubisco-N content per unit leaf area ($N_{rubisco}$), and the predicted cell wall N content per unit leaf area ($N_{structure}$) (Fig. 3). *A priori* we would expect the regression coefficient for $N_{structure}$ to be close to unity, and that for $N_{rubisco}$ to be about 6 to 20 (if Rubisco constitutes about 5 to 15% of total leaf protein: Evans 1989; Evans and Seemann 1989; Onoda et al. 2004). The fitted slopes of 1.2 ($p < 0.001$; 95% confidence limits: 1.0, 1.4) and 9.5 ($p < 0.001$; 7.6, 11.5) in Table 3, respectively, were consistent with these expectations.

There was no significant main effect of the factor 'N-fixer', and no significant interaction between $N_{rubisco}$ and the factor 'N-fixer'. The co-ordination hypothesis predicts that the metabolic component of $N_{area}$ should be environmentally optimized, and therefore independent of N supply. This could not be tested without direct measurements of $V_{cmax}$ or $N_{rubisco}$, which were precluded by the design of this study. However, N-fixers showed a steeper relationship between $N_{area}$ and $N_{structure}$. This was manifested as a significant interaction between the factor 'N-fixer' and $N_{structure}$ ($p < 0.01$). This model, in which $N_{area}$ was decomposed into a metabolic component predicted by the co-ordination hypothesis and a structural component proportional to LMA, explained 52% of the variance in $N_{area}$ across species and sites. The relative importance of variations in the metabolic and structural components, were determined to be 39% and 61% respectively, showing *inter alia* the importance of variation in LMA in determining leaf N content.

### 3.3 Quantifying trait plasticity *versus* species turnover

In total, 243 $C_3$ species were sampled at two or more sites. These species allowed calculation of a common slope, being an estimate of trait plasticity *sensu lato* (that is, phenotypic plasticity or genetic adaptation or both) across species (Fig. 4), for the traits $c_i{:}c_a$, ln LMA and ln $N_{area}$. Contrasting results were obtained for the three traits. It appeared that $c_i{:}c_a$ is perfectly plastic, with a common

(within-species) slope indistinguishable from unity. The common slope of $N_{area}$ was close to 0.5, indicating approximately equal contributions of plasticity and species turnover to the total variation. In the case of LMA, however, there was significant heterogeneity ($p < 0.05$) among the within-species slopes, with *Marsdenia viridiflora* showing a significantly steeper slope than the other species. After excluding this species, the common slope for LMA was also close to 0.5. A positive common slope indicates the ability of species to adapt their leaf morphology to environment. The positive common slope found for $N_{area}$ is consistent with this trait's nature as a combination of metabolic and structural components; its similarity to the slope for LMA is consistent with the importance of variations in structural N in determining total N.

## 4 Discussion

### 4.1 Leaf N and environment

The variety of environments provided in this study by the long transcontinental transect, and the number of species sampled, allowed us to statistically separate the effects of $c_i$:$c_a$, irradiance, temperature and LMA on $N_{area}$. The relationships to $c_i$:$c_a$, irradiance and temperature were in the directions and magnitudes predicted by the co-ordination hypothesis. The relationship to site mean irradiance had a slope as predicted by the co-ordination hypothesis (i.e. close to 1) but a strong relationship, with a steeper slope as expected, was found when top-of-canopy irradiance was used instead of the canopy mean – indicating that both spatial variations and within-canopy shading were contributing to the relationship with site mean irradiance. We performed an additional regression using leaf nitrogen content per unit mass ($N_{mass}$) which showed, as expected, identical fitted coefficients for all predictors except LMA (Appendix B: Table B1 and Fig. B1). However, because the regression coefficient of ln $N_{area}$ with respect to ln LMA < 1, the regression coefficient of ln $N_{mass}$ with respect to ln LMA < 0, i.e. $N_{mass}$ declines with increasing LMA – as has been widely reported. We also tried a regression of $N_{mass}$ on the same set of predictors but without the inclusion of LMA; this yielded a much poorer fit and is not shown.

High $N_{area}$ in plants from arid environments has been described often, and has traditionally been explained as a consequence of high N supply in environments with low rainfall (reducing leaching losses) and restricted plant cover (reducing total vegetation N demand) (e.g. Field and Mooney 1986).

This explanation would imply that plants in wetter environments have lower (and suboptimal) $N_{area}$ due to low *availability* of N. However, the negative relationship commonly found between $c_i:c_a$ and $N_{area}$ supports an alternative, adaptive (plant-centred) explanation. The least-cost hypothesis (Wright et al. 2003; Prentice et al. 2014) predicts lower $c_i:c_a$ in drier environments. This is because the drier the atmosphere, the greater the flux of water required to support a given rate of assimilation; which in turn shifts the balance of costs and benefits towards investment in photosynthetic capacity ($V_{cmax}$) and away from water transport capacity. When $c_i:c_a$ is lower, the co-ordination hypothesis predicts that a higher $V_{cmax}$ (and therefore higher $N_{area}$) is optimal, in order for the leaves to fully utilize the available light. The co-ordination hypothesis also predicts a further increase in $N_{area}$ with increasing aridity due to reduced cloudiness and reduced shading by competitors, both factors tending to increase $I_L$ (and both apparently contributing to the fitted relationship of $N_{area}$ to $I_L$). Thus the co-ordination hypothesis could account for independent positive effects of site irradiance and aridity on $N_{area}$, as previously reported by Wright et al. (2005). The fitted relationship of $N_{area}$ to temperature, PAR and $c_i:c_a$ is consistent with our theoretical prediction, which implicitly includes all of these effects.

Despite the large within-site variation in LMA found at all points along the aridity gradient, there is a significant tendency for LMA to increase with aridity, perhaps because of the resistance to dehydration conferred by stiffer leaves (Niinemets 2001; Wright and Westoby 2002; Harrison et al. 2010), and/or the need for leaves to avoid overheating under transient conditions of high radiation load and low transpiration rates combined with low wind speed (Leigh et al. 2012). This increase in LMA is inevitably accompanied by an increasing structural N component.

Thus, several distinct aspects of plant allocation tend to increase $N_{area}$ along gradients of increasing dryness. The predicted response of $N_{rubisco}$ to temperature is a result of opposing effects: the declining efficiency of photosynthesis with increasing temperature (due to the temperature dependencies of $K$ and $\Gamma^*$) is offset by the increased catalytic capacity of Rubisco at higher temperatures. The latter effect is predicted to be stronger, implying reduced $N_{area}$ with increasing temperature, as observed.

**4.2 The predictability of leaf N**

Predicted $N_{rubisco}$ and $N_{structure}$ together explained more than half of the variation in total $N_{area}$ across species and sites. Our approach to predicting these two quantities invokes a simplified formula, eq (5),

which is based on the co-ordination hypothesis for $N_{rubisco}$, assuming proportionality with Rubisco capacity; and assumes a simple proportionality with LMA for $N_{structure}$. Our finding of highly significant multiple regression coefficients for both variables indicates that the prediction obtained when taking both into account is more accurate than could be obtained from either variable alone. Osnas et al. (2013), analysing a large global leaf-trait data set and applying a novel method to determine the extent to which different traits are area- versus mass-proportional, found leaf N to be an intermediate case. This is to be expected if leaf N is, as our results suggest, a composite of an area-proportional ($N_{Rubisco}$) and a mass-proportional ($N_{structure}$) component. The two predictors (Rubisco capacity and LMA) are not fully independent, because leaves with higher photosynthetic capacity tend to have higher LMA for structural reasons. But such leaves must have increased structural N as well. By showing independently significant regression coefficients for modelled $N_{Rubisco}$ and LMA, the multiple regression results establish that successful prediction of $N_{area}$ requires consideration of both components; and that each has an independent effect, irrespective of their correlation ($r^2 = 0.28$ in this data set). Osnas et al. (2013) also fitted various statistical models for the relationships among leaf traits. Their 'model LN' for ln $N_{area}$ versus ln LMA yielded a slope of 0.38 (95% confidence interval 0.36 to 0.40). This value, based on a global data set, can be compared directly with − and is indistinguishable from − our fitted partial regression coefficient of ln Narea versus ln LMA, which is 0.42 (0.34 to 0.49) (Table 1).

In reality, however, leaf N does not consist exclusively of Rubisco and cell wall constituents. Leaf N includes multiple additional components including other photosynthetic proteins, proteins of the light-harvesting complexes and electron transport chains, cytosolic proteins, ribosomes and mitochondria, nucleic acids (which account for about 10–15% of leaf N: Chapin III and Kedrowski 1983), and N-based defensive compounds. It is possible that the higher N found for N-fixers resides in N-based osmolytes (Erskine et al. 1996) or defence compounds (Gutschick 1981). Nonetheless, our simplifications suggest that $N_{area}$ – especially at the community level, which is key for large-scale modelling – is, to first order, inherently predictable from leaf morphology and the physical environment. A corollary is that limitation in N supply may act primarily by changing plant allocation patterns (reducing allocation to light capture by leaves, while increasing allocation to N uptake by roots), rather than by altering leaf stoichiometry.

### 4.3 Trait variations within and between species

By testing for acclimation along spatial gradients, the design of our study did not allow phenotypic plasticity to be distingsuished from genetic adaptation. Phenotypic plasticity is the ability of a genotype to alter its expressed trait values in response to environmental conditions (Bradshaw 1965; Sultan 2000). A part of the observed variation in trait values within species could be due to shifts in the occurrence and frequency of different genotypes, producing different preferred trait values. Thus, when we refer to traits as 'plastic' this should be understood in a broad sense to allow the possibility of a genetic component of the observed adaptive differentiation within species. Seasonal acclimation within individual plants can provide more direct evidence for phenotypic plasticity (Togashi *et al.*, in revision), whereas in this study we disregard possible seasonal variations and instead relate trait variations to the mean annual environment. However, by sampling all of the species present at each site and including measurements on species at multiple sites, we could distinguish between the contribution of plasticity *sensu lato* (phenotypic plasticity and/or genetic adaptation) versus species turnover, i.e. the progressive replacement of species with different mean trait values, to spatial variation in the community mean values of a given trait. We found that $\delta^{13}C$ was perfectly plastic, perhaps not surprisingly as variations in $c_i{:}c_a$ are under stomatal control. In contrast, LMA and $N_{area}$ showed approximately equal contributions from plasticity and species turnover.

## 4.4 Implications for modelling

There has been a surge of interest in schemes to predict continuous trait variation in DGVMs (e.g. Scheiter et al. 2013; Fyllas et al. 2014; van Bodegom et al., 2014; Ali et al. 2015; Fisher et al. 2015; Meng et al. 2015; Sakschewski et al. 2015). Some trait-based modelling approaches have relied on empirical information on trait-trait and trait-environment covariation, but others (e.g. Scheiter et al. 2013) have aimed to represent the adaptive nature of trait variation explicitly. Our focus has been on testing an explicit adaptive hypothesis for the controls of one key trait, $N_{area}$, which in addition to a structural component (necessarily linked to LMA) includes an important metabolic component, reflecting the leaf-level investment in photosynthetic proteins. All models that attempt to represent the coupling between C and N cycles in terrestrial ecosystems require a method to calculate leaf N content, given other environmental and plant characteristics. Some models prescribe fixed values of $V_{cmax}$ (per plant functional type) but this approach does not take account of the observed variation in $V_{cmax}$ with environmental conditions. Models that assume proportionality between $V_{cmax}$ and $N_{area}$ neglect the

important variation in leaf structural N. We have shown that $N_{area}$ is predictable, to a degree that is useful for modelling, when both metabolic and structural components are taken into account. Our prediction is based on LMA, $c_i$:$c_a$ and a *theoretically predicted* value of $V_{cmax}$ based on the co-ordination hypothesis – for which there is strong independent evidence (e.g. Maire et al. 2012). The partial responses of $N_{area}$ to $c_i$:$c_a$, irradiance and temperature are consistent with predictions of the co-ordination hypothesis, and the inclusion of predicted $V_{cmax}$ adds significantly and substantially to the predictive power of LMA and $c_i$:$c_a$ alone. As both LMA (Wright et al. 2005) and $c_i$:$c_a$ (Prentice et al. 2014) show relationships to environment, our results suggest a possible route towards a general adaptive scheme for the prediction of major leaf traits in DGVMs, which would be an improvement on models that assume a one-to-one relationship between photosynthetic capacity and $N_{area}$ (see e.g. Adams et al. 2016, who showed that there is considerable variation in $N_{area}$ among N-fixers that is unrelated to photosynthetic capacity). Our results also suggest some priorities for trait data collection and analysis: to test the predicted controls of $N_{area}$ over a wider range of environments, and to test the predicted environmental controls of $V_{cmax}$ directly in the field.

Our application of trait gradient analysis also points out a way towards process-based treatments of functional trait diversity in next-generation models. It is increasingly accepted that models could, and should, sample 'species' from continuous gradients of traits rather than fixing the traits associated with discrete PFTs. A hybrid approach to modelling $N_{area}$ based on the present analysis would consider $N_{area}$ explicitly as the sum of metabolic and structural components. The metabolic component would be treated as plastic, and subject to environmental optimization (in space and time) consistent with the least-cost and co-ordination hypotheses. The structural component would be tied to LMA, which is a key variable of the 'leaf economics spectrum' (Wright et al. 2004), strongly expressed both within and between environments and therefore requiring a broad range of values to be assigned to model 'species'.

Finally, we note that if our results can be corroborated more widely, this would point to the need for a shift in the way N 'limitation' is treated – both in models and in analyses of field data. In studies of the relationship between $V_{cmax}$ and leaf N, for example, it is conventional to plot N on the x-axis and $V_{cmax}$ on the y-axis, and it is then often stated that the positive relationship found shows that variation in leaf N 'causes' variation in $V_{cmax}$. But all that is shown on the graph is a correlation, and our 'plant-centred'

interpretation is the opposite of the conventional one: that is, $V_{cmax}$ is adaptively matched (acclimated) to environmental conditions, and the metabolic component of leaf N is a consequence of this acclimation. Low N availability would then result in reduced allocation of C (and N) to leaves, and increased allocation below ground – which is also an adaptive response, but at the whole-plant rather than the leaf level.

## Appendices

### Appendix A: Theoretical responses of $N_{area}$ to environmental predictors

We estimate optimal $V_{cmax}$ by $\varphi_0\, I_L\, (c_i + K)/(c_i + 2\Gamma^*)$ (eq 5). Holding other variables constant, the sensitivity of this estimate to absorbed PAR is given by the derivative of its natural logarithm with respect to $\ln I_L$:

$$\partial \ln V_{cmax} / \partial \ln I_L \quad = 1 \tag{A1}$$

Similarly, the sensitivity of this estimate to $c_i$ is given by:

$$\partial \ln V_{cmax}/\partial c_i \quad = \quad (2\Gamma^* - K)/[(c_i + K)(c_i + 2\Gamma^*)] \tag{A2}$$

and its sensitivity to the $c_i{:}c_a$ ratio is smaller than this by a factor $c_a$.

Temperature-dependent reaction rates are described by the Arrhenius equation:

$$\ln x\,(T) - \ln x\,(T_{ref}) \quad = \quad (\Delta H/R)\,(1/T_{ref} - 1/T) \tag{A3}$$

where $x$ is the rate parameter of interest, $T$ is the measurement temperature (K), $T_{ref}$ is the reference temperature (here 298 K), $\Delta H$ is the activation energy of the reaction (J mol$^{-1}$ K$^{-1}$) and $R$ is the universal gas constant (8.314 J mol$^{-1}$ K$^{-1}$). Linearizing eq (A3) around $T_{ref}$ yields:

$$\ln x\,(T) - \ln x\,(T_{ref}) \quad \approx \quad (\Delta H/RT_{ref}^2)\,\Delta T \tag{A4}$$

where $\Delta T = T - T_{ref}$. Thus, from equation (5):

$$\ln V_{cmax25} \quad \approx \quad \ln V_{cmax} \quad - \quad (\Delta H_v/RT_{ref}^2)\,\Delta T \tag{A5}$$

where $\Delta H_v$ is the activation energy of $V_{cmax}$. The sensitivity of $V_{cmax25}$ to $T$ is then:

$$\partial \ln V_{cmax25}/\partial T \quad = \quad \partial \ln V_{cmax}/\partial T - (\Delta H_v/RT_{ref}^2)$$
$$= \quad (\partial K/\partial T)/(c_i + K) - 2(\partial \Gamma^*/\partial T)/(c_i + 2\Gamma^*) - (\Delta H_v R/T_{ref}^2) \tag{A6}$$

where $K = K_c\,(1 + O/K_o)$, hence:

$$\partial K/\partial T = \partial K_c/\partial T + [(\partial K_c/\partial T)\,K_o - (\partial K_o/\partial T)\,K_c]\,O/K_o^2 \tag{A7}$$

where $O$ is the atmospheric concentration of oxygen and $\Gamma^*$ and the Michaelis-Menten coefficients for carboxylation ($K_C$) and oxygenation ($K_O$) respectively have values at $T_{ref}$ (in µmol mol$^{-1}$) and activation energies as given by Bernacchi *et al.* (2001).

## Appendix B: Partial responses of $N_{mass}$ to environmental predictors

**Table B1**. Linear regression coefficients for ln ($N_{mass}*100$) (g g$^{-1}$) as a function of $c_i{:}c_a$ (from $\delta^{13}$C), ln (mean canopy PAR, $I_L$) (µmol m$^{-2}$ s$^{-1}$), MAT (°C), ln LMA (g m$^{-2}$) and the factor 'N-fixer' at species level. Note $N_{mass}$ was multiplied by 100 before logarithmic transformation

|  | Estimated | Predicted | $p$ | $R^2$ |
|---|---|---|---|---|
| $c_i{:}c_a$ | $-0.611 \pm 0.252$ | $-0.615$ | <0.01 | |
| ln $I_L$ | $0.874 \pm 0.096$ | 1 | <0.001 | |
| MAT | $-0.047 \pm 0.007$ | $-0.048$ | <0.001 | 51% |
| ln LMA | $-0.585 \pm 0.036$ | n/a | <0.001 | |
| 'N-fixer' | $0.306 \pm 0.041$ | n/a | <0.001 | |

**Fig B1**. Partial residual plots for the regression of ln ($N_{mass} \times 100$) (g g$^{-1}$) as a function of $c_i{:}c_a$ (from $\delta^{13}$C), ln (mean canopy PAR, $I_L$) (µmol m$^{-2}$ s$^{-1}$), MAT (°C), ln LMA (g m$^{-2}$) and the factor 'N-fixer' at species level.

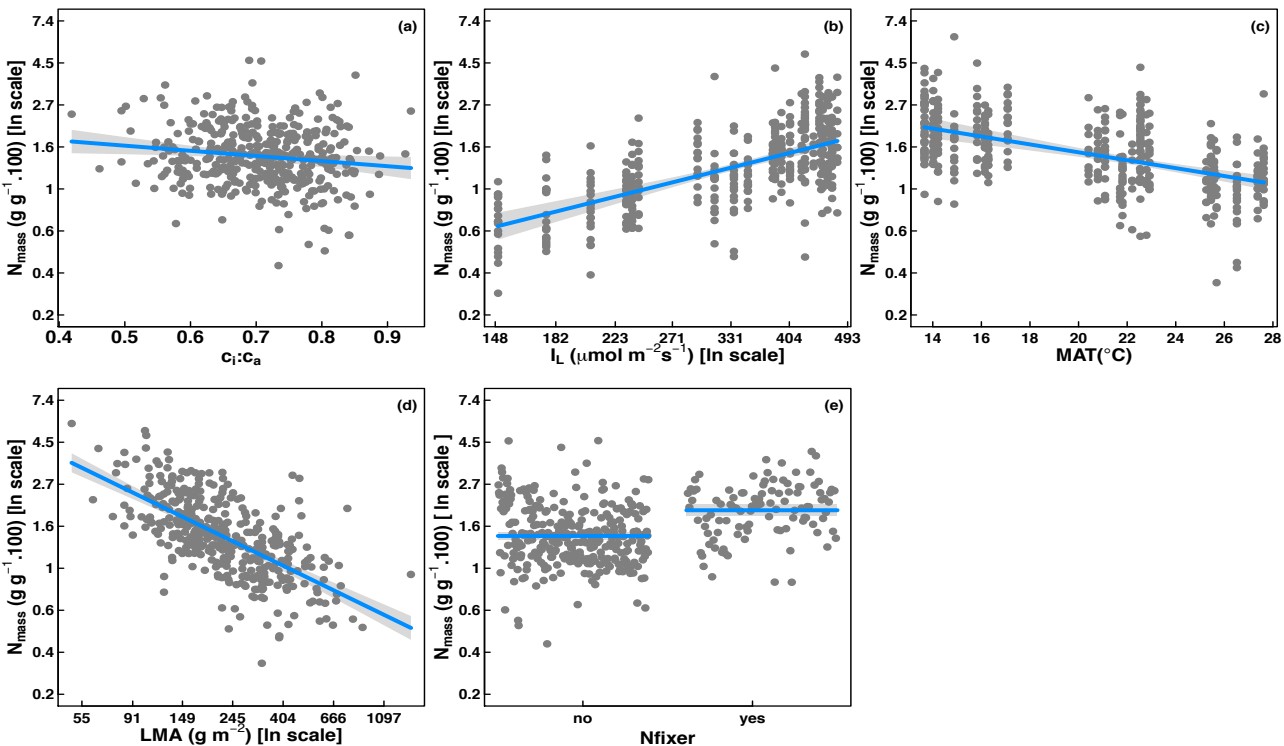

**Author contribution**

ICP, ND and AJL planed and designed the study; ND carried out all the field measurement and performed the data analyses. ND and ICP wrote the first draft; BJE supported the study through provision of climate data; IJW assisted with data interpretation, contributed with ideas throughout and suggested important improvements to the text. SCR contributed important ideas to improve text. All authors contributed on subsequent versions.

**Acknowledgments**

Research funded by the Terrestrial Ecosystem Research Network (TERN) through the AusPlots, Australian Transect Network and eMAST facilities (http://www.emast.org.au). DN was supported by an international Macquarie University Research Scholarship and eMAST facilities. IJW has been supported by an Australian Research Council Future Fellowship (FT100100910). BJE have been supported by eMAST. Thanks to the AusPlots Rangelands team (particularly Emrys Leitch, Christina Pahl and Ben Sparrow) for undertaking field work and detailed consultation; Rosemary Taplin, Ian fox, Peter Latz and Emrys Leitch for plant identification; Belinda Medlyn for insisting that the assumptions in the LPJ model must be tested; Yusuke Onoda for providing the empirical relationship between LMA and cell-wall N. Discussions with Yan-Shih Lin and Han Wang helped to improve the data analysis. This work is a contribution to the AXA Chair Programme in Biosphere and Climate Impacts and the Imperial College Initiative on Grand Challenges in Ecosystems and the Environment.

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

**Table 1**. Linear regression coefficients for ln $N_{area}$ (g m$^{-2}$) as a function of $c_i{:}c_a$ (from $\delta^{13}$C), ln (mean canopy PAR, $I_L$) (μmol m$^{-2}$ s$^{-1}$), MAT (˚C), ln LMA (g m$^{-2}$) and the factor 'N-fixer' at species level.

|  | Estimated | Predicted | $p$ | Relative importance | $R^2$ |
|---|---|---|---|---|---|
| $c_i{:}c_a$ | −0.611 ± 0.252 | −0.615 | <0.01 | 14% | |
| ln $I_L$ | 0.874 ± 0.096 | 1 | <0.001 | 19% | |
| MAT | −0.047 ± 0.007 | −0.048 | <0.001 | 9% | 55% |
| ln LMA | 0.415 ± 0.036 | n/a | <0.001 | 39% | |
| 'N-fixer' | 0.306 ± 0.041 | n/a | <0.001 | 19% | |

**Table 2**. Linear regression coefficients for community-mean (simple average) values of ln $N_{area}$ (g m$^{-2}$) as a function of $c_i{:}c_a$ (from $\delta^{13}$C), ln (mean canopy PAR, $I_L$) (µmol m$^{-2}$ s$^{-1}$), MAT (˚C) and ln LMA (g m$^{-2}$).

| | Estimated | Predicted | $p$ | Relative importance | $R^2$ |
|---|---|---|---|---|---|
| $c_i{:}c_a$ | −1.60 ± 0.94 | −0.615 | n.s. | 42% | |
| ln $I_L$ | 0.70 ± 0.23 | 1 | <0.001 | 20% | |
| MAT | −0.035 ± 0.016 | −0.048 | <0.001 | 11% | 82% |
| ln LMA | 0.57 ± 0.19 | n/a | <0.001 | 27% | |

**Table 3**. Linear regression coefficients for $N_{area}$ as a function of independently predicted values of $N_{rubisco}$ and $N_{structure}$ (all in g m$^{-2}$) at species level.

| | Estimated | Predicted | $p$ | Relative importance | $R^2$ |
|---|---|---|---|---|---|
| $N_{rubsico}$ | 9.5 ±2.0 | 6-20 | <0.001 | 39% | |
| $N_{structure}$ | 1.2 ± 0.2 | 1 | <0.001 | 61% | 52% |
| $N_{structure}$:*'N-fixer'* | 1.0 ± 0.3 | n/a | <0.01 | n/a | |

**Figures.**

**Fig 1** Site locations, climate and leaf trait distributions: Mean annual precipitation (MAP, mm), mean annual temperature (MAT, ˚C), mean incident daytime photosynthetically active radiation (PAR, μmol m$^{-2}$ s$^{-1}$), moisture index (MI). Site mean $N_{area}$ (g m$^{-2}$) and LMA (g m$^{-2}$) are also shown.

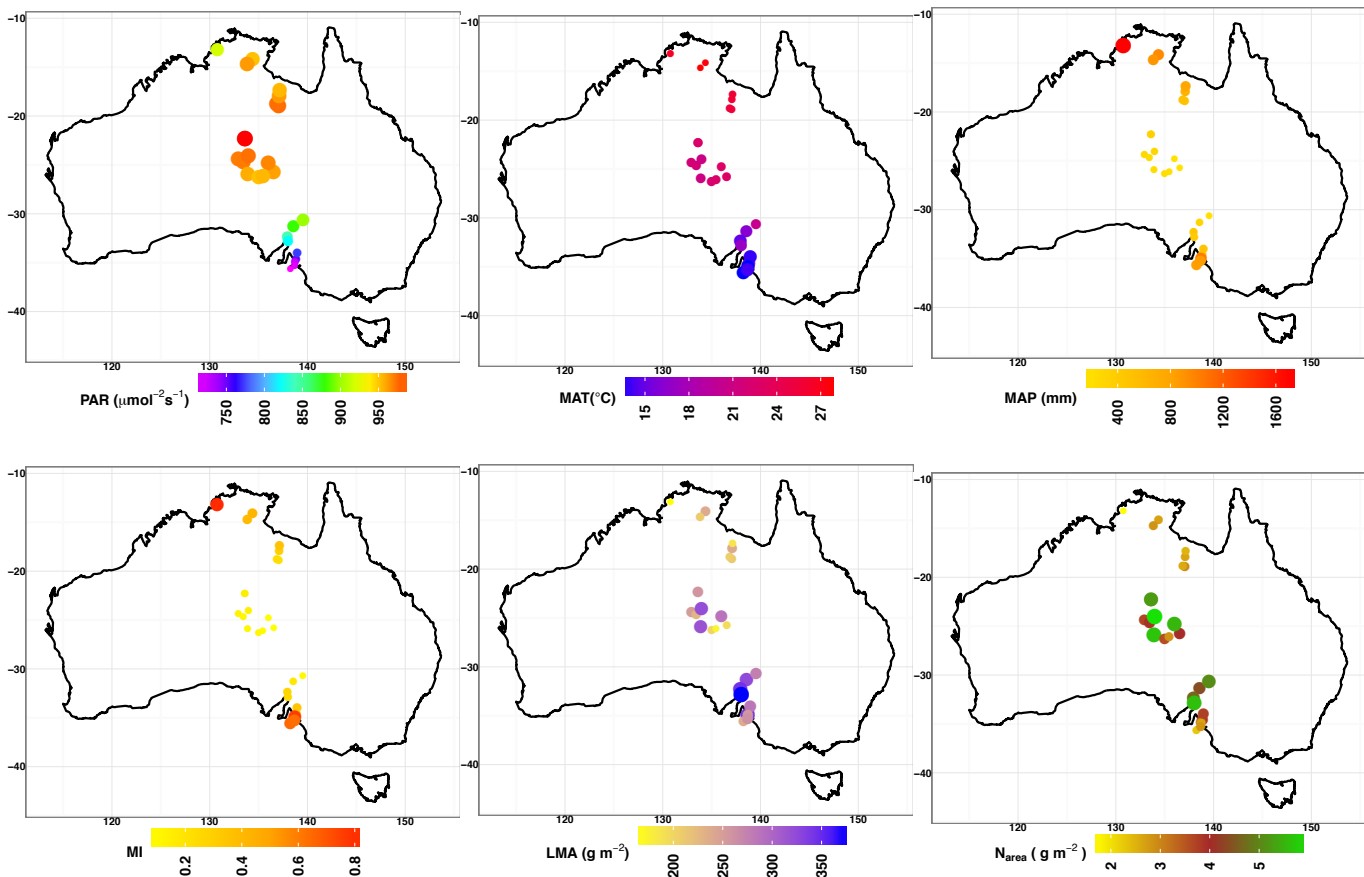

**Fig 2**. Partial residual plots for the regression of ln $N_{area}$ (g m$^{-2}$) as a function of $c_i{:}c_a$ (from $\delta^{13}C$), ln (mean canopy PAR, $I_L$) ($\mu$mol m$^{-2}$ s$^{-1}$), MAT (˚C), ln LMA (g m$^{-2}$) and the factor 'N-fixer' at species level. Note the logarithmic scale of the y-axis.

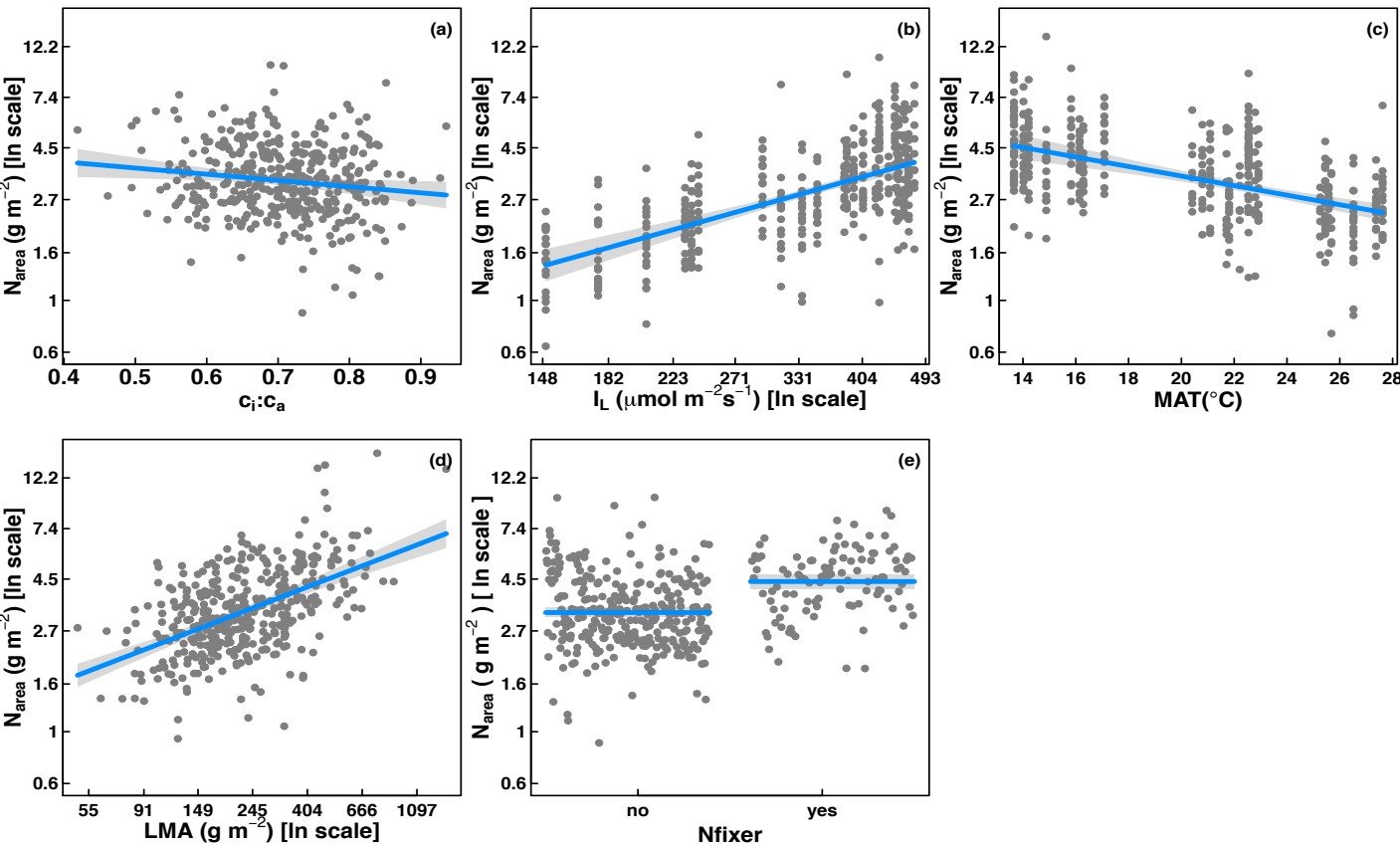

**Fig 3**. Partial residual plots for the linear regression of $N_{area}$ as a function of independently predicted values of $N_{rubisco}$ and $N_{structure}$ (all in g m$^{-2}$) at species level. Blue: N-fixers, red: non-N-fixers.

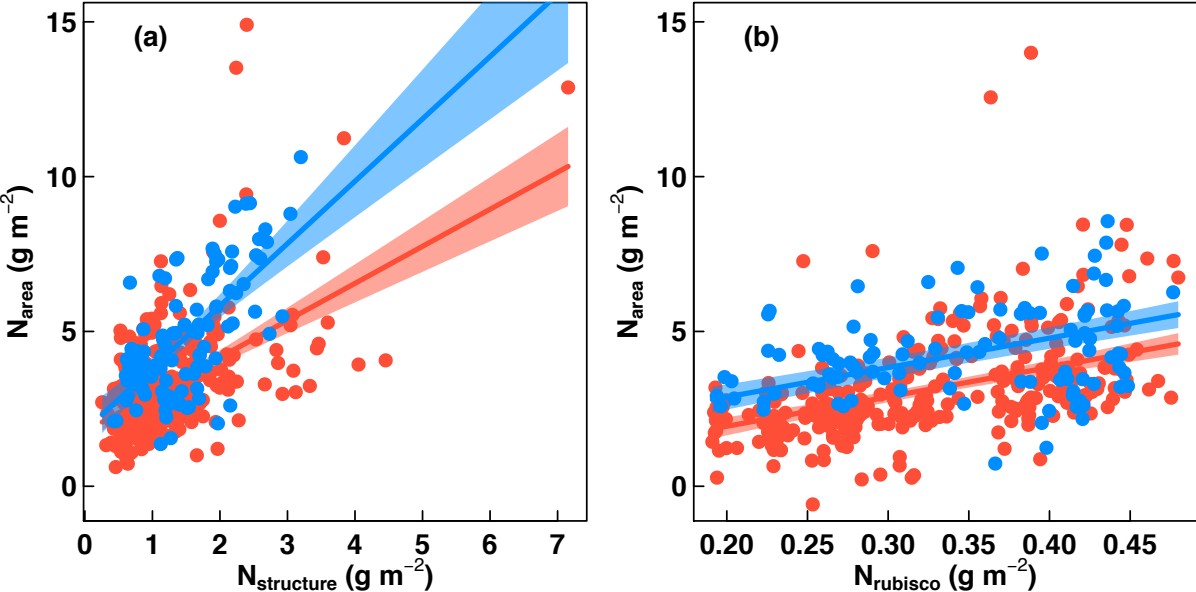

**Fig 4**. Trait means and regression lines for all 243 $C_3$ plant species in the 27 study sites. Note the logarithmic scales for $N_{area}$ (g m$^{-2}$) and LMA (g m$^{-2}$). Thin red dashed lines represent individual within-species regression lines of non-N-fixer species. Thin blue lines represent individual within-species regression lines of N-fixer species. The black dashed line represents the overall regression line, which has a slope of unity by definition. Grey dots denote individual species-site combinations. Common within-species slopes are $0.53 \pm 0.11$ (ln $N_{area}$), $1.02 \pm 0.12$ ($c_i$:$c_a$) and $0.55 \pm 0.11$ (ln LMA)

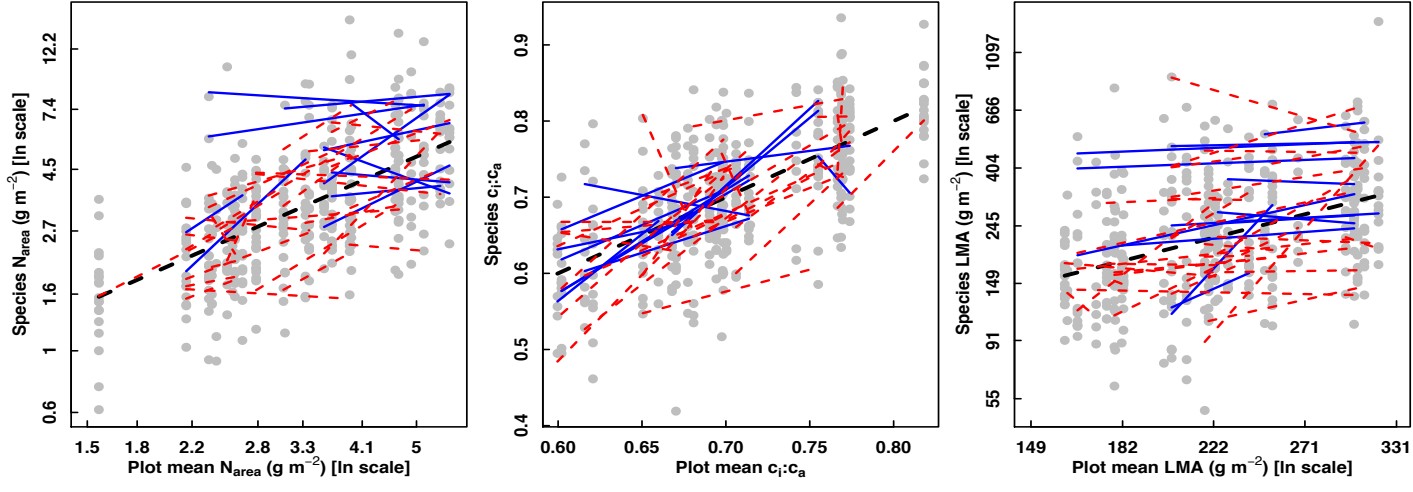