# Peer review of "Leaf nitrogen from first principles: field evidence for adaptive variation with climate"

_Biogeosciences, 2016_

## Short Comment (SC1) · 15 Apr 2016

I found this paper very interesting, but did have two quick suggestions:

1. The authors write:

"For example, any modelling approach that predicts photosynthetic capacity from Narea, and Narea in turn from soil inorganic N supply (Luo et al. 2004), is incompatible with the hypothesis that photosynthetic capacity is optimized at the leaf level as a function of irradiance, leaf internal $CO_2$ concentration ($c_i$) and temperature (Haxeltine and Prentice 1996, Dewar 1996) – as assumed in the widely used LPJ DGVM (Sitch et al. 2003) and other models derived from it, including LPJ GUESS (Smith et al. 2001) and LPX (Prentice et al. 2011a; Stocker et al. 2013)."

[Figure]

I wonder if this could be explained a little further? I think it is an important point, but don't feel that it is immediately self evident why these hypotheses cannot co-exist, i.e. that a canopy can optimise for leaf N, but be constrained by supply from the soil inorganic N, e.g. McMurtrie et al. 2008, Functional Plant Biology, 2008, 35, 521-534. I think it

2. Fig 1:

Remove the labels from the points and increase their size. Currently you cannot see the colour variation very easily.

---

## Referee Comment (RC1) · Anonymous Referee #1 · 28 Apr 2016

This paper presents an analysis of how leaf N per area (Na) varies with climate in terms of its structural and functional (photosynthetic) components. The effects are attributed to inter-specific variation and within species adaptation, and the results are interpreted in a leaf optimization framework.

I find this study to be extraordinary in going all the way from leaf sampling to modeling, producing empirical evidence, theoretical progress, and new components for predictive models in one paper. It is a rarely seen example of how to combine observations and theory to make real quantitative progress, beyond the usual "significant or not" testing of ecological hypotheses. In conclusion, I find this an very useful contribution to the research area.

I only have some minor questions/suggestions: In the discussion, p.10 l. 19, the least

cost hypothesis is explained as reducing ci/ca in drier environment due to the need for increased water transport capacity for a given rate of assimilation. Why this happens is not obvious to me. I would have thought that in drier environments water limitation would force the plants to increase water use efficiency by increasing assimilation capacity (Amax) per water use? Maybe an additional line of explanation could help here.

Then in the final comments it is suggested that Vcmax should be plotted on the X axis against leaf N instead of the usual opposite way. I think I get the point of this, but at the same time, isn't N in proteins a key part of the machinery or structure that performs the "function Vcmax". I think both ways of plotting could be equally valid also from an plant centered perspective.
* * *

---

## Referee Comment (RC2) · Anonymous Referee #2 · 11 May 2016

This study sets out to predict leaf nitrogen per unit area (Narea) through a combination of leaf mass per unit area (LMA), the ratio of leaf-internal to atmospheric co2 (ci:ca) and Rubisco activity.

Although the study presents some interesting observations relating environmental variables to Narea and other leaf-scale traits, a major omission has been made by not showing explicitly how nitrogen per unit leaf mass (Nmass) varies in these observations. It is possible to infer some aspects of the relationships from the data presented, but it seems possible that a much simpler and perhaps stronger predictive relationship could be formulated around the simple fact that Narea = LMA * Nmass. This relationship is clear to the authors as they use it to calculate Narea itself from measurements of LMA and Nmass (p.5 line 24).

[Figure]

The authors attempt to separate the LMA contribution to variation in Narea from a metabolic contribution, but they arrive at a summation of effects, one connected to structural variation which is tightly connected to LMA, and another metabolic component that is formulated as independent of LMA (p.2 lines 12-14, p.7 lines 4-6). My concern with this approach is that the metabolic component of Narea includes a dependence on LMA as well, since metabolic variation can be driven both by changes in the leaf tissue N concentration and by the number of layers of mesophyll cells and the thickness of each layer.

Without explicitly showing how Nmass is related to the environmental factors explored here, it is not clear how the current study moves the field forward from the relationship suggested by Niinemets and Tenhunen (1997) between Vcmax and Narea.

There is also a potential incongruency in the calculation of irradiance as a function of canopy leaf area, while asserting that the leaves measured were from the sunlit canopy. If truly sunlit leaves werre used, then the relevant irradiance would be the top of canopy values. Perhaps this is just a matter of defining what sunlit leaves means for species that exist only in the understory of mixed species canopies. In any case, I am concerned that the irradiance used for sunlit leaves of the dominant trees in these relationships is not the correct one.

―――――――――――――――――――――――――――――

---

## Author Comment (AC1) · 26 Jun 2016

**Comments from M. G. De Kauwe (mdekauwe@gmail.com)**

I found this paper very interesting, but did have two quick suggestions:

"For example, any modelling approach that predicts photosynthetic capacity from Narea, and Narea in turn from soil inorganic N supply (Luo et al. 2004), is incompatible with the hypothesis that photosynthetic capacity is optimized at the leaf level as a function of irradiance, leaf internal $CO_2$ concentration ($c_i$) and temperature (Haxeltine and Prentice 1996, Dewar 1996) – as assumed in the widely used LPJ DGVM (Sitch et al. 2003) and other models derived from it, including LPJ GUESS (Smith et al. 2001) and LPX (Prentice et al. 2011a; Stocker et al. 2013)." I wonder if this could be explained a little further? I think it is an important point, but don't feel that it is immediately self evident why these hypotheses cannot co-exist, i.e. that a canopy can optimise for leaf N, but be constrained by supply from the soil inorganic N, e.g. McMurtrie et al. 2008, Functional Plant Biology, 2008, 35, 521-534...

*Response:* At the leaf level, the co-ordination hypothesis predicts that photosynthetic capacity is optimized as a function of irradiance, leaf internal $CO_2$ concentration ($c_i$) and temperature. At the whole plant level, we expect limited N supply to be manifested in a limitation on canopy size (i.e. number of leaves) rather than on the photosynthetic capacity of the individual leaves. We propose to state this more explicitly in a revised version

*"For example, one modelling approach predicts photosynthetic capacity from $N_{area}$, and $N_{area}$ in turn from soil inorganic N supply (e.g. Luo et al. 2004). But this is incompatible with the hypothesis that photosynthetic capacity is optimized at the leaf level as a function of irradiance, leaf-internal $CO_2$ concentration ($c_i$) and temperature (Haxeltine and Prentice 1996, Dewar 1996) – as assumed in the widely used LPJ DGVM (Sitch et al. 2003) and other models derived from it, including LPJ-GUESS (Smith et al. 2001) and LPX (Prentice et al. 2011a; Stocker et al. 2013). **This 'plant-centred' hypothesis is based on the idea that plant allocation processes***

*determine leaf-level traits. Limited N supply, by this reasoning, should lead to the production of fewer leaves, rather than leaves with suboptimal capacity."*

2. Fig 1:

Remove the labels from the points and increase their size. Currently you cannot see the colour variation very easily.

*Response:* We have made an improved version of Fig 1, following this suggestion.

The revised Fig 1 is as follows:

**Fig 1** Site locations, climate and leaf trait distributions: Mean annual precipitation (MAP, mm), mean annual temperature (MAT, ˚C), mean incident daytime photosynthetically active radiation (PAR, μmol m$^{-2}$ s$^{-1}$), moisture index (MI). Site mean $N_{area}$ (g m$^{-2}$) and LMA (g m$^{-2}$) are also shown.

[Figure]

---

## Author Comment (AC2) · 26 Jun 2016

**Comments from Anonymous Referee #1**

This paper presents an analysis of how leaf N per area (Na) varies with climate in terms of its structural and functional (photosynthetic) components. The effects are attributed to inter-specific variation and within species adaptation, and the results are interpreted in a leaf optimization framework. I find this study to be extraordinary in going all the way from leaf sampling to modeling, producing empirical evidence, theoretical progress, and new components for predictive models in one paper. It is a rarely seen example of how to combine observations and theory to make real quantitative progress, beyond the usual "significant or not" testing of ecological hypotheses. In conclusion, I find this an very useful contribution to the research area.

*Response:* we thank the referee for this appreciative comment.

In the discussion, p.10 l. 19, the least cost hypothesis is explained as reducing ci/ca in drier environment due to the need for increased water transport capacity for a given rate of assimilation. Why this happens is not obvious to me. I would have thought that in drier environments water limitation would force the plants to increase water use efficiency by increasing assimilation capacity (Amax) per water use? Maybe an additional line of explanation could help here.

*Response:* nitrogen and water use are substitutable for each other, according to the least-cost hypothesis. The marginal cost of water transport is relatively high, compared to that of nitrogen use, in drier environments. Thus, plants adapt to a drier environment by reducing water loss by adopting a lower $c_i/c_a$ ratio, while increasing carbon fixation capacity ($V_{cmax}$).

This reasoning follows from an economic perspective on plant adaptation, which replaces the concept of 'limitation' with the concept of relative costs and benefits. However, the outcome is fully consistent with the referee's suggestion, because lower $c_i/c_a$ implies increased water use efficiency (assimilation per unit water use).

To make this clearer, we propose to amend the text as follows:

*"The least-cost hypothesis (Wright et al. 2003; Prentice et al. 2014) predicts lower $c_i:c_a$ in drier environments. This is because the drier the atmosphere, the greater the flux of water required to support a given rate of assimilation; which in turn shifts the balance of costs and benefits towards investment in photosynthetic capacity ($V_{cmax}$) and away from water transport capacity."*

Then in the final comments it is suggested that Vcmax should be plotted on the X axis against leaf N instead of the usual opposite way. I think I get the point of this, but at the same time, isn't N in proteins a key part of the machinery or structure that performs the "function Vcmax". I think both ways of plotting could be equally valid also from an plant centered perspective.

***Response:*** N in proteins is indeed a key part of the machinery that supports $V_{cmax}$. However, we think our slightly provocative point is worth making, because when the two variables are plotted in the usual way, it is easy to infer (as is very commonly done, in both empirical and modelling studies) that the N content of the leaf is an independent 'cause' of its $V_{cmax}$. We are suggesting the reverse: that the metabolic N content of the leaf is a consequence of the $V_{cmax}$ adopted by the leaf.

It is also too easy, and incorrect, to regard leaf N simply as a proxy for $V_{cmax}$. As our analysis suggested, the non-photosynthetic component of leaf N can be large in plants with high LMA. Adams et al. (2016) have shown that photosynthetic capacity is not related to $N_{area}$ in N-fixing plants, and pointed out that leaf N can also perform other functions such as defence against herbivory. There is also evidence that the predominant effect of artificially increasing N availability by fertilizer addition is increased canopy size (see e.g. Rosati et al., 2000 and references therein). Although a full review would be beyond the scope of this paper, there seem to be several lines of evidence suggesting that the common modelling approach, whereby N supply regulates photosynthetic capacity, needs replacing.

We propose to expand our discussion of this point, and to refer to Adams et al. (2016) in support of our argument and revised the text as followed in the discussion:

*"As both LMA (Wright et al. 2005) and $c_i{:}c_a$ (Prentice et al. 2014) show relationships to environment, our results suggest a possible route towards a general adaptive scheme for the prediction of major leaf traits in DGVMs, which would be an improvement on models that assume a one-to-one relationship between photosynthetic capacity and $N_{area}$ (see e.g. Adams et al. 2016, who showed that there is considerable variation in $N_{area}$ among N-fixers that is unrelated to photosynthetic capacity)."*

**References:**

Adams, M. A., Turnbull, T. L., Sprent, J. I., and Buchmann, N.: Legumes are different: Leaf nitrogen, photosynthesis, and water use efficiency, Proceedings of the National Academy of Sciences, 113, 4098-4103, 2016.

Rosati, A., Day, K. R., and DeJong, T. M.: Distribution of leaf mass per unit area and leaf nitrogen concentration determine partitioning of leaf nitrogen within tree canopies, Tree Physiology, 20, 271-276, 2000.

---

## Author Comment (AC3) · 26 Jun 2016

Comments from anonymous Referee #2

This study sets out to predict leaf nitrogen per unit area (Narea) through a combination of leaf mass per unit area (LMA), the ratio of leaf-internal to atmospheric $CO_2$ (ci:ca) and Rubisco activity. Although the study presents some interesting observations relating environmental variables to Narea and other leaf-scale traits, a major omission has been made by not showing explicitly how nitrogen per unit leaf mass (Nmass) varies in these observations. It is possible to infer some aspects of the relationships from the data presented, but it seems possible that a much simpler and perhaps stronger predictive relationship could be formulated around the simple fact that Narea = LMA * Nmass. This relationship is clear to the authors as they use it to calculate Narea itself from measurements of LMA and $N_{mass}$ (p.5 line 24).

***Response:*** We have carried out an analysis of ln $N_{mass}$, parallel to our analysis of ln $N_{area}$, and propose to present this in an Appendix. But since the relationship between ln $N_{mass}$ and ln $N_{area}$ can be expressed by ln $N_{area}$ = ln $LMA$ + ln $N_{mass}$, the results are predictable: the partial relationships to variables other than ln LMA are unchanged, while the regression coefficient of $N_{mass}$ with respect to ln LMA is reduced by 1. Because the coefficient of $N_{area}$ with respect to ln LMA < 1, the coefficient of $N_{mass}$ with respect to ln LMA < 0 (i.e. $N_{mass}$ declines with LMA). But this relationship is neither simpler of stronger than our main analysis.

We also tried an analysis of $N_{mass}$ omitting LMA as a predictor, but this resulted in a much poorer fit with several non-significant coefficients. We propose to add some explanation of these additional results in the discussion.

*"We also performed a parallel regression using leaf nitrogen content per unit mass ($N_{mass}$) which showed, as expected, identical fitted coefficients for all predictors except LMA (Appendix B: Table B1 and Fig. B1). However, because the regression coefficient of ln $N_{area}$ with respect to ln LMA < 1, the regression coefficient of ln $N_{mass}$*

*with respect to ln LMA < 0, i.e. N$_{mass}$ declines with increasing LMA, as has been widely reported. We also tried a regression of N$_{mass}$ on the same set of predictors but without the inclusion of LMA; this yielded a much poorer fit and is not shown."*

The authors attempt to separate the LMA contribution to variation in Narea from a metabolic contribution, but they arrive at a summation of effects, one connected to structural variation which is tightly connected to LMA, and another metabolic component that is formulated as independent of LMA (p.2 lines 12-14, p.7 lines 4-6). My concern with this approach is that the metabolic component of Narea includes a dependence on LMA as well, since metabolic variation can be driven both by changes in the leaf tissue N concentration and by the number of layers of mesophyll cells and the thickness of each layer.

**Response:** We independently predict the structural and metabolic components of leaf N. The structural component of leaf N is assumed to be proportional to LMA, and this assumption is supported by an independent analysis of the relationship between cell-wall N and LMA (see p. 11, line 14). The metabolic component of leaf N is assumed to be proportional to $V_{cmax}$ at a given temperature, which is predicted as a function of irradiance, leaf-internal $CO_2$ concentration ($c_i$) and temperature.

Now in reality, as the referee notes, $V_{cmas}$ is not entirely independent of LMA, because leaves with high $V_{cmax}$ require high LMA. But this means they require more structural N as well. Our multiple regression approach remains valid, even if LMA and $V_{cmax}$ are partially correlated; the fact that we obtain independently significant regression coefficients indicates that both make separate contributions to determining $N_{area}$.

We propose to add a sentence to (a) recognize the partial dependence of LMA on $V_{cmax}$ and (b) note how this is handled by multiple regressions in the discussion.

*"These two predictors are not fully independent, because leaves with higher photosynthetic capacity tend to have higher LMA for structural reasons. But such leaves must have increased structural N as well. By showing independently significant*

*regression coefficients for modelled $N_{Rubisco}$ and LMA, the multiple regression approach establishes that successful prediction of $N_{area}$ requires consideration of both components."*

Without explicitly showing how Nmass is related to the environmental factors explored here, it is not clear how the current study moves the field forward from the relationship suggested by Niinemets and Tenhunen (1997) between Vcmax and Narea.

***Response:*** The proposed revision will include a demonstration of the partial relationships between $N_{mass}$ and environmental variables, as mentioned above.

Niinemets and Tenhunen (1997) is an important reference for this research. Their focus was on explaining the observed vertical gradients of photosynthetic capacity and $N_{area}$ within a tree canopy. Our focus is on predicting observed patterns in $N_{area}$ across species and environments. Our success in doing so represents a significant advance on earlier work.

There is also a potential incongruency in the calculation of irradiance as a function of canopy leaf area, while asserting that the leaves measured were from the sunlit canopy. If truly sunlit leaves were used, then the relevant irradiance would be the top of canopy values. Perhaps this is just a matter of defining what sunlit leaves means for species that exist only in the understory of mixed species canopies. In any case, I am concerned that the irradiance used for sunlit leaves of the dominant trees in these relationships is not the correct one.

***Response:*** Our terminology was wrong: we should have referred to 'outer canopy' leaves rather than 'sunlit' leaves, and we propose to amend this in the revision.

*"Mature **outer-canopy** leaves were sampled during the growing season using the AusPlots methodology (White et al. 2012)."*

By calculating a canopy-average irradiance, we represent the conditions likely to be experienced by species on average. This will indeed underestimate the irradiance experienced by the outer leaves of dominant trees or shrubs. It will also overestimate the irradiance experienced by plants at ground level. These errors presumably contribute to the scatter around the fitted relationship of $N_{area}$ with irradiance. We add some words of explanation on this point in the method.

*"In dense vegetation $I_L$ will underestimate the PAR exposure of canopy dominants and overestimate the PAR exposure of understory species. However, the use of a canopy average in this way was a necessary approximation (because we did not have quantitative information about the canopy position of each species) and considered preferable to using $I_0$, which will systematically overestimate PAR exposure for most species in a dense community."*

**Appendix B: Partial responses of $N_{mass}$ to environmental predictors**

**Table B1**. Linear regression coefficients for ln $N_{mass}$ (g g$^{-1}$). 100 as a function of $c_i:c_a$ (from $\delta^{13}$C), ln (mean canopy PAR, $I_L$) ($\mu$mol m$^{-2}$ s$^{-1}$), MAT (˚C), ln LMA (g m$^{-2}$) and the factor 'N-fixer' at species level. Note $N_{mass}$ was multiplied by 100 before log transformation.

| | Estimated | Predicted | $p$ | $R^2$ |
|---|---|---|---|---|
| $c_i:c_a$ | −0.611 ± 0.252 | −0.615 | <0.01 | |
| ln $I_L$ | 0.874 ± 0.096 | 1 | <0.001 | |
| MAT | −0.047 ± 0.007 | −0.048 | <0.001 | 51% |
| ln LMA | -0.585 ± 0.036 | n/a | <0.001 | |
| 'N-fixer' | 0.306 ± 0.041 | n/a | <0.001 | |

**Fig B1**. Partial residual plots for the regression of ln $N_{mass}$ (g g$^{-1}$). 100 as a function of $c_i$:$c_a$ (from $\delta^{13}$C), ln (mean canopy PAR, $I_L$) (mmol m$^{-2}$ s$^{-1}$), MAT (℃), ln LMA (g m$^{-2}$) and the factor 'N-fixer' at species level. Note $N_{mass}$ was multiplied by 100 before logarithmic transformation.

[Figure]

---

## Author Response (AR1)

**Consolidated response to comments**

**Editor:**

Based on your responses to the two reviewers I invite you to prepare a revised manuscript. Further to the discussion on the effects of Narea versus Nmass and the role of LMA I recommend you to consider also the paper by Osnas et al. (2013) published in Science.

*This is an excellent point.*

*Osnas et al. (2013) performed an extensive analysis on a large global leaf trait data set. Their principal aim was to resolve the question of whether quantitative leaf traits should most appropriately be expressed on an area or a mass basis. In doing so, they fitted various statistical models that are relevant to our MS. Their 'Model LN' is particularly of interest. This can be written as:*

*$\ln N_{area}(k) = I' + S' \ln LMA(k) + n'(k)$*

*where (k) refers to an individual measurement, n'(k) is a zero-mean normal random variable, and I' and S' are constants to be estimated. They estimated a value for S' of 0.38 (95% confidence interval 0.36 to 0.40), which is statistically indistinguishable from our estimate of the OLS slope of $\ln N_{area}$ versus $\ln$ LMA, viz. 0.42 (0.34 to 0.49). We have added text to this effect (page 12, lines 10-14):*

*"Osnas et al. (2013) also fitted various statistical models for the relationships among leaf traits. Their 'model LN' for $\ln N_{area}$ versus $\ln$ LMA yielded a slope of 0.38 (95% confidence interval 0.36 to 0.40). This value, based on a global data set, can be compared directly with – and is indistinguishable from – our fitted partial regression coefficient of ln Narea versus ln LMA, which is 0.42 (0.34 to 0.49) (Table 1)."*

*Osnas et al. (2013) also used a novel approach to determine the extent to which each quantitative leaf trait could be considered to be area- or mass-dependent. Whereas some traits were unambiguously one or the other, leaf N fell in between – consistent with our analysis indicating that leaf N can be broken down into a component proportional to leaf area, and a component proportional to leaf mass. We have now commented on this consistency between our findings and those of Osnas et al. (2013) (page 12, lines 2-6):*

*"Osnas et al. (2013), analysing a large global leaf-trait data set and applying a novel method to determine the extent to which different traits are area- versus mass-proportional, found leaf N to be an intermediate case. This is to be expected if leaf N is, as our results suggest, a composite of an area-proportional ($N_{Rubisco}$) and a mass-proportional ($N_{structure}$) component."*

**Anonymous Referee #1**

This paper presents an analysis of how leaf N per area (Na) varies with climate in terms of its structural and functional (photosynthetic) components. The effects are attributed to inter-specific variation and within species adaptation, and the results are interpreted in a leaf optimization framework. I find this study to be extraordinary in going all the way from leaf sampling to modeling, producing empirical evidence, theoretical progress, and new components for predictive models in one paper. It is a rarely seen example of how to combine observations and theory to make real quantitative progress, beyond the usual "significant or not" testing of ecological hypotheses. In conclusion, I find this an very useful contribution to the research area.

*We thank the referee for this appreciative comment.*

In the discussion, p.10 l. 19, the least cost hypothesis is explained as reducing ci/ca in drier environment due to the need for increased water transport capacity for a given rate of assimilation. Why this happens is not obvious to me. I would have thought that in drier environments water limitation would force the plants to increase water use efficiency by increasing assimilation capacity (Amax) per water use? Maybe an additional line of explanation could help here.

*Nitrogen and water are substitutable resources according to the least-cost hypothesis. Thus, a plant can invest in additional photosynthetic capacity while closing stomata, reducing the requirement for water transport. Or it can invest in additional water transport capacity, allowing more open stomata, while economizing on photosynthetic capacity. The same assimilation rate is achieved either way. In drier environments, the marginal cost of water transport is relatively high, compared to that of nitrogen use; so the optimal strategy is to adopt a lower $c_i/c_a$ ratio, while increasing $V_{cmax}$. This outcome is fully consistent with the referee's suggestion, because lower $c_i/c_a$ implies increased water use efficiency (assimilation per unit water use).*

*To make this clearer, we have amended the text as follows (page 11, lines 5-8):*

[The least-cost hypothesis (Wright et al. 2003; Prentice et al. 2014) predicts lower $c_i:c_a$ in drier environments]. *This is because the drier the atmosphere, the greater the flux of water required to support a given rate of assimilation; which in turn shifts the balance of costs and benefits towards investment in photosynthetic capacity ($V_{cmax}$) and away from water transport capacity."*

Then in the final comments it is suggested that Vcmax should be plotted on the X axis against leaf N instead of the usual opposite way. I think I get the point of this, but at the same time, isn't N in proteins a key part of the machinery or structure that performs the "function Vcmax". I think both ways of plotting could be equally valid also from an plant centered perspective.

*N in proteins is indeed a key part of the machinery that supports $V_{cmax}$. However, we think our slightly provocative point is worth making, because when the two variables are plotted in the usual way, it is easy to infer (as is very commonly done, in both empirical and modelling studies) that the N content of the leaf is a* **cause** *of its $V_{cmax}$. We are suggesting the reverse: that the metabolic N content of the leaf is a* **consequence** *of the $V_{cmax}$ adopted by the leaf.*

*It is also too easy, and incorrect, to regard leaf N simply as a proxy for $V_{cmax}$. As our analysis suggests, the non-photosynthetic component of leaf N can be large in plants with high LMA. Adams et al. (2016) have shown that photosynthetic capacity is not related to $N_{area}$ in N-fixing plants, and pointed out that leaf N can also perform other functions than photosynthesis (such as defence against herbivory). There is also evidence that the predominant effect of artificially increasing N availability by fertilizer addition is increased canopy size (see e.g. Rosati et al., 2000 and references therein). Although a fuller review would be beyond the scope of this paper, there seem to be several lines of evidence suggesting that the common modelling approach, whereby N supply regulates photosynthetic capacity, needs replacing.*

*In support of our argument, we have added the following text (page 13 line 29; page 14 lines 1-3):*

[our results suggest a possible route towards a general adaptive scheme for the prediction of major leaf traits in DGVMs,] *which would be an improvement on models that assume a one-to-one relationship between photosynthetic capacity and $N_{area}$ (see e.g. Adams et al. 2016, who showed that there is considerable variation in $N_{area}$ among N-fixers that is unrelated to photosynthetic capacity).*

**Anonymous Referee #2**

This study sets out to predict leaf nitrogen per unit area (Narea) through a combination of leaf mass per unit area (LMA), the ratio of leaf-internal to atmospheric CO2 (ci:ca) and Rubisco activity. Although the study presents some interesting observations relating environmental variables to Narea and other leaf-scale traits, a major omission has been made by not showing explicitly how nitrogen per unit leaf mass (Nmass) varies in these observations. It is possible to infer some aspects of the relationships from the data presented, but it seems possible that a much simpler and perhaps stronger predictive relationship could be formulated around the simple fact that Narea = LMA * Nmass. This relationship is clear to the authors as they use it to calculate Narea itself from measurements of LMA and $N_{mass}$ (p.5 line 24).

*In response to this proposal, we have carried out an analysis of ln $N_{mass}$, parallel to our analysis of ln $N_{area}$, and have presented this in Table B1 and Fig. B1 in the new Appendix B. But since the relationship between ln $N_{mass}$ and ln $N_{area}$ can be expressed by ln $N_{area}$ = ln LMA + ln $N_{mass}$, the results were predictable: the partial relationships to variables other than ln LMA are unchanged, while the regression coefficient of*

*$N_{mass}$ with respect to ln LMA is reduced by exactly 1. Because the coefficient of ln $N_{area}$ with respect to ln LMA < 1, the coefficient of ln $N_{mass}$ with respect to ln LMA < 0 (i.e. $N_{mass}$ declines with LMA). But this relationship is neither simpler, nor stronger, than our main analysis based on $N_{area}$.*

*We also tried an analysis of $N_{mass}$ omitting LMA as a predictor, but this resulted in a much poorer fit with several non-significant coefficients. We have added some explanation of these additional results. See page 10, lines 20-26:*

*"We performed an additional regression using leaf nitrogen content per unit mass ($N_{mass}$) which showed, as expected, identical fitted coefficients for all predictors except LMA (Appendix B: Table B1 and Fig. B1). However, because the regression coefficient of ln $N_{area}$ with respect to ln LMA < 1, the regression coefficient of ln $N_{mass}$ with respect to ln LMA < 0, i.e. $N_{mass}$ declines with increasing LMA, as has been widely reported. We also tried a regression of $N_{mass}$ on the same set of predictors but without the inclusion of LMA; this yielded a much poorer fit and is not shown."*

The authors attempt to separate the LMA contribution to variation in Narea from a metabolic contribution, but they arrive at a summation of effects, one connected to structural variation which is tightly connected to LMA, and another metabolic component that is formulated as independent of LMA (p.2 lines 12-14, p.7 lines 4-6). My concern with this approach is that the metabolic component of Narea includes a dependence on LMA as well, since metabolic variation can be driven both by changes in the leaf tissue N concentration and by the number of layers of mesophyll cells and the thickness of each layer.

*We independently predict the structural and metabolic components of leaf N. The structural component of leaf N is assumed to be proportional to LMA, and this assumption is supported by an independent analysis of the relationship between cell-wall N and LMA (see p. 11, line 14). The metabolic component of leaf N is assumed to be proportional to $V_{cmax}$ at a given temperature, which is predicted as a function of irradiance, leaf-internal $CO_2$ concentration ($c_i$) and temperature.*

*Now in reality, as the referee notes, $V_{cmax}$ is not entirely independent of LMA, because leaves with high $V_{cmax}$ require high LMA. But this means they require more structural N as well. Our multiple regression approach remains valid, even if LMA and $V_{cmax}$ are partially correlated. The fact that we obtain independently significant regression coefficients indicates that both make separate, significant contributions to determining $N_{area}$.*

*We have added new text (a) to recognize the partial dependence of LMA on $V_{cmax}$ and (b) to note how this is handled by multiple regression, as follows (page 12, lines 6-10):*

*"The two predictors (Rubisco capacity and LMA) are not fully independent, because leaves with higher photosynthetic capacity tend to have higher LMA for structural*

*reasons. But such leaves must have increased structural N as well. By showing independently significant regression coefficients for modelled $N_{Rubisco}$ and LMA, the multiple regression approach establishes that successful prediction of $N_{area}$ requires consideration of both components."*

Without explicitly showing how Nmass is related to the environmental factors explored here, it is not clear how the current study moves the field forward from the relationship suggested by Niinemets and Tenhunen (1997) between Vcmax and Narea.

*First, the revised text includes a demonstration of the partial relationships between $N_{mass}$ and environmental variables (Table B1 and Fig. B1 in Appendix B).*

*Second, although Niinemets and Tenhunen (1997) is an important reference for this research, their focus was on explaining the observed vertical gradients of photosynthetic capacity and $N_{area}$. Our focus is on predicting observed patterns in $N_{area}$ more broadly, across species and environments. Our success in doing so therefore represents a significant advance on this earlier work.*

There is also a potential incongruency in the calculation of irradiance as a function of canopy leaf area, while asserting that the leaves measured were from the sunlit canopy. If truly sunlit leaves were used, then the relevant irradiance would be the top of canopy values. Perhaps this is just a matter of defining what sunlit leaves means for species that exist only in the understory of mixed species canopies. In any case, I am concerned that the irradiance used for sunlit leaves of the dominant trees in these relationships is not the correct one.

*Our terminology was wrong: we should have referred to 'outer canopy' leaves rather than 'sunlit' leaves! We have amended this in the revision (page 5, lines 26).*

*By calculating a canopy-average irradiance, we represent the conditions likely to be experienced by species **on average**. This will indeed underestimate the irradiance experienced by the outer leaves of the canopy dominants, but it will also overestimate the irradiance experienced by plants at ground level. Such errors presumably contribute to the scatter around the fitted relationship of $N_{area}$ with irradiance. We have added some words of explanation on this point (page 5, lines 19-24):*

*"In dense vegetation $I_L$ will underestimate the PAR exposure of canopy dominants and overestimate the PAR exposure of understory species. However, the use of a canopy average in this way was a necessary approximation (because we did not have quantitative information about the canopy position of each species) and considered preferable to using $I_0$, which will systematically overestimate PAR exposure for most species in a dense community."*

**M. G. De Kauwe** (mdekauwe@gmail.com)

I found this paper very interesting, but did have two quick suggestions:

"For example, any modelling approach that predicts photosynthetic capacity from Narea, and Narea in turn from soil inorganic N supply (Luo et al. 2004), is incompatible with the hypothesis that photosynthetic capacity is optimized at the leaf level as a function of irradiance, leaf internal CO2 concentration (ci) and temperature (Haxeltine and Prentice 1996, Dewar 1996) – as assumed in the widely used LPJ DGVM (Sitch et al. 2003) and other models derived from it, including LPJ GUESS (Smith et al. 2001) and LPX (Prentice et al. 2011a; Stocker et al. 2013)." I wonder if this could be explained a little further? I think it is an important point, but don't feel that it is immediately self evident why these hypotheses cannot co-exist, i.e. that a canopy can optimise for leaf N, but be constrained by supply from the soil inorganic N, e.g. McMurtrie et al. 2008, Functional Plant Biology, 2008, 35, 521-534.

*At the **leaf** level, the co-ordination hypothesis predicts that photosynthetic capacity is optimized as a function of irradiance, leaf internal $CO_2$ concentration ($c_i$) and temperature. At the **whole plant** level, however, we expect limited N supply to be manifested in a limitation on canopy size (i.e. number of leaves) rather than on the photosynthetic capacity of the individual leaves. This has now been stated explicitly in the revised text (page 3, limes 20-21):*

*"(Limited N supply, by this reasoning, should lead to the production of fewer leaves, rather than leaves with suboptimal capacity.)"*

Fig 1: Remove the labels from the points and increase their size. Currently you cannot see the colour variation very easily.

*Done and replaced it with the revised Fig.1 (page 26).*

**Literature cited in this document:**

Adams, M. A., Turnbull, T. L., Sprent, J. I., and Buchmann, N.: Legumes are different: Leaf nitrogen, photosynthesis, and water use efficiency, Proceedings of the National Academy of Sciences, 113, 4098-4103, 2016.

Osnas, J. L. D., Lichstein, J. W., Reich, P. B., and Pacala, S. W.: Global leaf trait relationships: mass, area, and the leaf economics spectrum, Science, 340, 741-744, 2013.

Rosati, A., Day, K. R., and DeJong, T. M.: Distribution of leaf mass per unit area and leaf nitrogen concentration determine partitioning of leaf nitrogen within tree canopies, Tree Physiology, 20, 271-276, 2000.

[revised manuscript text omitted]

---

## Author Response (AR2)

**Consolidated response to comments**

**Editor:**

I apologize for the unexpected delay in the handling of your manuscript, which was due to the fact that the more critical reviewer, who wanted to re-assess your revised manuscript, repeatedly missed the deadline (partly because you had forgotten to include Appendix B in the manuscript). For this reason I have invited another reviewer, who was very supportive of your paper, but pointed out a remaining inconsistency concerning the canopy position of sampled leaves versus your estimate of absorbed PAR. I agree that this is critical, as part of your analysis is based on the relationship of these two parameters. When submitting a revised manuscript please add a cover letter explaining how you have addressed all points raised by the reviewer and make sure to include Appendix B and the associated Table and Figure this time.

*Apologies for this oversight. We have now included Appendix B, with the associated Table and Figure.*

**Response to Anonymous Referee #3 (Report 2)**

This manuscript provides an outstanding combination of plant eco-physiological theory and empirical data. I support its publication.

*Thank you for your support!*

However, I have one request and a few minor suggestions, which may improve clarity of the text and information content.

Apart from this, I suggest to make the original trait measurements available, or - if the data are already available - mention, where the data are available.

*As these data have not yet been published in a repository, we have provided them (both the leaf traits and the ancillary variables) in the form of a Supplement.*

Page 5: The description of the leaf sampling strategy and the calculation of respective absorbed PAR on community level seem not consistent: while absorbed PAR represents the canopy average, leaves for analyses have been selected as: 'mature outer canopy leaves'. This inconsistency has already been criticized in the earlier version. To me the formulation in the current version is still either unfortunate, or there is indeed an inconsistency in the analyses, or there is an aspect involved that I do not yet understand. I think, this needs at least clarification.

*There is no inconsistency, but our wording was still unclear. In denser vegetation (woodland in this study), many species sampled are in the understorey and therefore*

*not "sunlit". When for example an understorey shrub was sampled, the sampled leaves were indeed taken from the outer canopy of the shrub – but still in the shade of the overstorey. We have further clarified this point by adding a clarification, as follows (in page 6, lines 17):*

(Note that in denser vegetation many species sampled are in the understorey, so their 'outer-canopy' leaves are still shaded by the overstorey. Many species thus receive considerably reduced sunlight compared to the overstorey, implying that the canopy-average irradiance $I_L$ is more suitable than the top-of-canopy value $I_0$ as a community measure of irradiance.)

Minor suggestions:

- Table 1-3: I suggest to decompose contributions of individual drivers (as has been done for table 3, but only in the text).

*We have provided this information now in all three Tables in the revised manuscript.*

- Figure 3: I suggest to use the same scale on the 2 y-axes. I assume that Narea values should be the same in both plots a and b?

*Figure 3 has been revised as suggested, using the same scale for the two y-axes:*

**Fig 3**. Partial residual plots for the linear regression of $N_{area}$ as a function of independently predicted values of $N_{rubisco}$ and $N_{structure}$ (all in g m$^{-2}$) at species level. Blue: N-fixers, red: non-N-fixers.

[Figure]

- Page 3 Line 20/21: I suggest to skip the brackets.

*Done*

- Page 4: I think providing some overview of analyses at the beginning of Material and Methods would help readers. I therefore suggest to first provide an overview over analyses and data collected and then provide details on the individual methods and data. The overview could be adapted from the description of analyses on page 7.

*An overview of the analyses has now been provided in the Material and Methods section, as follows (in page 5, lines 7):*

Our analyses are based on 442 leaf measurements representing all species found in a 100 m ×100 m plot at each of 27 sites on a broad North-South transect across Australia (Fig. 1) We performed a regression analysis to test the relationships of $N_{area}$ to mean annual temperature (MAT), irradiance, plant traits leaf mass per area (LMA), $c_i{:}c_a$ ratio and N-fixation capacity. We also fitted a statistical model in which $N_{area}$ was treated as the sum of a metabolic component proportional to predicted (optimal) photosynthetic capacity at standard temperature (based on temperature, irradiance and $c_i{:}c_a$ ratio) and a structural component proportional to LMA. Finally, we carried out a trait gradient analysis in order to quantify the contributions of environment versus species identity to variation in $N_{area}$, $c_i{:}c_a$ ratio and LMA.

- Equation 5: Based on the coordination hypothesis Vcmax was calculated from ci:ca and MAT, using some additional photosynthesis parameters. Where do the values for K and gamma_star come from? Has this equation be tested empirically? If so, the reference would be appreciated.

*Yes, these are well established. Values of K and Γ\* at a reference temperature of 25˚C, and their activation energies which determine how they vary with temperature, are empirically determined* in vivo *values as provided by Bernacchi et al. (2001). We have revised the text to make this clear, by adding the following statement (in page 7, lines 14):*

Values of both these quantities and their activation energies (governing their temperature responses) are based on the empirical *in vivo* determinations by Bernacchi et al. (2001), widely used in photosynthesis research.

**Response to late review**

I appreciate the efforts made by the authors to address the previous review comments. I still have some concerns about the broad purpose of the paper as reflected in the title, abstract, and introduction as compared to the unique and valuable results obtained.

*With this comment the reviewer seems very positive overall, but makes a number of points below, some of them repeated from the previous review, which we find to be (although closely argued) misdirected for the most part.*

*In several cases we have now proposed additional sentences and modified wording intended to discourage readers from similar misconceptions. These, we suspect, arise from the fact that our work actually has some surprisingly radical implications both for ecosystem modelling and for the interpretation of trait data – turning some widespread assumptions on their heads. There are many ramifications arising from our 'plant-centred' framework that we have not attempted to spell out here, because (a) further evidence would be required (we have plenty that is still unpublished) and (b) we do wish now to publish these strong results, which the framework has inspired us to generate.*

Major comments:

1. My main concern is that both the title, with its wording "from first principles", and the abstract (second paragraph) suggest that a major contribution from this work is to provide a mechanistic prediction of *Narea* that would be useful in ecosystem models. Since the approach described here requires knowledge of ci:ca (estimated from d13C), LMA, and leaf area index (to estimate canopy mean irradiance), it is not a useful framework for a mechanistic ecosystem model. If all of those properties are assumed known, then it is not clear why a model would need further information about *Narea* itself.

***Any** model that tries to represent the coupling between N and C cycling needs to keep track of the amount of N in the leaves (and other tissues, but we do not consider these here). The reviewer suggests that a knowledge of $c_i:c_a$ ratio, LMA and LAI provides all the information required to predict $N_{area}$. This is not so, and we are unsure why the reviewer seems so certain about it. Models have to invoke additional assumptions: such as that $N_{area}$ is a function of N supply from the soil, and/or that it is proportional to $V_{cmax}$ (which many models prescribe as a fixed value for each plant functional type). Our results provide a way in which $N_{area}$ can indeed be predicted from those three quantities, without such questionable assumptions, and with a strong theoretical and empirical basis.*

*We have added a new sentence in the Abstract (page 2, lines 17) and two more in section 4.4(page 14, lines 26) which, we hope, will make it somewhat clearer how this study can indeed provide useful information for modellers.*

*Revised in abstract (page 2, lines 17)*

Coupled carbon-nitrogen models require a method to predict $N_{area}$ that is more realistic than the widespread assumptions that $N_{area}$ is proportional to photosynthetic capacity, and/or that $N_{area}$ (and photosynthetic capacity) are determined by N supply from the soil.

*Revised in section 4.4 (page 14, lines 26)*

All models that attempt to represent the coupling between C and N cycles in terrestrial ecosystems require a method to calculate leaf N content, given other environmental and plant characteristics. Some models prescribe fixed values of $V_{cmax}$ (per plant functional type) but this approach does not take account of the observed variation in $V_{cmax}$ with environmental conditions. Models that assume proportionality between $V_{cmax}$ and $N_{area}$ neglect the important variation in leaf structural N. [We have shown that $N_{area}$ is predictable, to a degree that is useful for modelling,] when both metabolic and structural components are taken into account.

- In addition, the quantitative framework offered, while useful for helping to understand variation in *Narea*, is not what I would call a "first principles" approach, but is rather much more an empirical approach the results of which offer support for a variety of hypotheses about mechanisms associated with plant trait variation. I think the results of the trait gradient analysis are unique and interesting, and I encourage the authors to reframe the title, abstract, and introduction to focus on this aspect of the work.

*The reviewer seems to propose a radical rewriting, in which the trait gradient analysis becomes the main focus of the paper. However, other reviewers have found merit in the rest of the paper, where our focus is on the predictability of leaf N. Our view is that both aspects are important.*

*Moreover, we would like to defend our use of the expression "first principles" because our study is **not** simply an empirical exploration of the determinants of $N_{area}$. There are many published plant-trait papers of this nature, which unlike ours, do not provide key new information for model development. Instead of simply seeking statistical relationships, here having first conceptualized $N_{area}$ as the sum of terms proportional to LMA and $V_{cmax}$, we estimate $V_{cmax}$ using a novel theoretical derivation based on the co-ordination hypothesis; and we show that parameters of our theoretical model estimated from the data are consistent with values predicted independently. Thus, we start from an **explicit theoretical basis** which represents both a novel approach to the analysis of trait data and, by extension, a fundamental departure from current modelling practice.*

*Thus we have not made the suggested major change, because it would go against the recommendations of the other reviewers; and we have not removed the reference to "first principles", as we consider it to this phrase to be an entirely appropriate*

*description of our analysis.*

2. Another concern is that the revised manuscript still does not adequately deal with the influence of LMA on Vcmax. The units of Vcmax are never expressed, but it seems clear that the manuscript is using the conventional approach of an area-based estimate. The approach used here of treating Narea as a linear combination of a "structural" and a "metabolic" component, and then considering LMA only in the "structural" part leaves the strong impression that variation in LMA is being accounted for in the structural part and is not contributing to further variation in the metabolic part. The manuscript indicates that this is correct to "first order", but I doubt that conclusion. The structural component is intended to capture the influence of additional cell wall material that necessarily accompanies higher LMA, based on Onoda's work. But of course, the area-based Vcmax is very strongly regulated by variation in LMA, particularly for variation within one species driven by different irradiance. This should be mentioned more explicitly in relation to the results in Section 3.2.

*$V_{cmax}$ is almost universally expressed per unit area (µmol $m^{-2}$ $s^{-1}$) and we followed that convention. We have now stated this explicitly in the Introduction (page 3, lines 7) so as to avoid any confusion about units as follow.*

[… the maximum rate of carboxylation ($V_{cmax}$) at standard temperature], also expressed per unit area

*We have removed the phrase "to first order" because this was only meant to hedge against the fact that leaf N includes additional components that are neither structural nor related to photosynthesis. But this matter is anyway dealt with in the Discussion. It seems this phrase suggested to the reviewer some hesitation on our part, but we do not hesitate to assert that the main components of leaf N are indeed structural (proportional to cell wall material) and metabolic (proportional to Rubisco). Indeed our results provide strong support for this concept. We re-iterate that the fact that LMA and $V_{cmax}$ are expected to be (partially) correlated does not invalidate our finding of independently significant regression coefficients for both predictors. Our analysis does not in any way deny the existence of this correlation. We have added explicit new text on this point in two places in the revised text: in the first paragraph of the Introduction (in page 3, lines 12), and in section 4.2(in page 13, lines 4;page 13, lines 14) as follow.*

*Revised in introduction (in page 3, lines 12):*

[Thus, $N_{area}$] can usefully be considered as [the sum of a 'metabolic' component related to $V_{cmax}$ and a 'structural' component proportional to LMA]. Leaves with high $V_{cmax}$ usually have high LMA and so these two quantities can be at least partially correlated, as seen clearly (for example) in parallel vertical gradients of $V_{cmax}$ and

LMA within canopies of one species (e.g. Niinemets and Tenhunen 1997). Across different species and environments, however, there is scope for considerable independent variation in $V_{cmax}$ and LMA, implying the need to consider them separately.

*Revised in section 4.2(in page 13, lines 4):*

Our finding of highly significant multiple regression coefficients for both variables indicates that the prediction obtained when taking both into account is more accurate than could be obtained from either variable alone.

*Revised in section 4.2(in page 13, lines 14):*

[…], and that each has an independent effect, irrespective of their correlation ($r^2 =$ 0.28 in this data set).

*The reviewer also misses a key point when it is stated that "**of course,** the area-based Vcmax is very strongly regulated by variation in LMA..." (**Bold** ours) and refers to the example of "variation with irradiance within a species driven by different irradiance". This key point is that the **correlation** between LMA and $V_{cmax}$ does not imply that LMA **causes** $V_{cmax}$. In our view, the high LMA of upper-canopy leaves in forests should instead be considered as a **consequence** of the optimal $V_{cmax}$ being high at high light, because high $V_{cmax}$ cannot be achieved otherwise. We hope this is now abundantly clear.*

3. The discussion of Nmass is improved, but still I think should be more explicit. It is clear that for any leaf, when two of the three (LMA, Narea, Nmass) are known the third is given. However, environmental drivers on variation in canopy function can drive variation in canopy N distribution that could show up as gradients in some, all, or none of these quantities. The Introduction suggests that these authors favor an interpretation of observations from nature in which Nmass is relatively constant while Narea varies, presumably through the influence of additional canopy layers and the well-studied light-driven gradient in LMA. I don't disagree with this interpretation, but I feel that the predictions for Nmass based on the coordination hypothesis and the attempt to linearize variation in Narea into structural and metabolic components should be made explicit.

*We are puzzled by this comment and cannot think of a suitable improvement to the text in order to address it. It is said that "the attempt to linearize variation in Narea into structural and metabolic components should be made explicit", but this is already explicit. Evidently this reviewer is interested in how our findings (across species and environments) relate to the better-studied vertical gradients within*

*canopies. We agree this would be of great interest but the issue is well beyond the scope of our paper, given that the design of our study did not allow us to gather data on within-canopy gradients. We suggest that the broad topic raised here would be more suitable for a different paper, either a review, or perhaps analysis of a different data set including both macroclimatic and microenviromental gradients*

4. The discussion of different modeling approaches or hypotheses given in the Introduction (p.3 l.10 through p.4 l.10) is very cursory and leaves much of the reasoning implicit. There is an effort to contrast models based on the co-ordination hypothesis with models that connect variation in leaf-scale N with N supply. The assertion is made that the two approaches are incompatible, but the rest of this paragraph does not attempt to back up that assertion, and instead focuses on the theoretical and empirical underpinnings of the co-ordination approach. It seems that this contrast between modeling approaches could be an interesting topic for a paper on its own, but the manuscript in its current form is not seriously considering how the observations do or do not support this comparison. This material either needs to be strengthened or the assertion removed. In either case I think the content in the latter part of this paragraph can stand on its own without reference to compatibility with other modeling approaches.

*Our treatment of this matter was indeed brief, and we are happy to strengthen it because it is an important part of the context of our analysis. We have therefore clarified our logic in the Introduction (page 3,lines 23), more specifically pointing out the basic incompatibility: either $N_{area}$ is principally controlled by soil microbial processes (which is the assumption behind one common approach to modelling photosynthetic capacity), or it is controlled by plant allocation processes, which is the key to our plant-centred approach. Both cannot be true at the same time.*

*Revised in introduction (page 3,lines 23)*

This implies an assumption that the soil environment, and soil microbial activity in particular, are the primary controls of $N_{area}$ and photosynthetic capacity at the leaf level. An alternative assumption [is that photosynthetic capacity is optimized…]

5. Another concern is related to the method of estimating $I_L$ and the impact of that methodology on the conclusions about the relationship with Narea. Since $I_L$ is a canopy mean which is influenced by the total canopy leaf area, variation in $I_L$ will be driven to a large (but here unquantified) degree by variation in leaf area index (LAI). In canopies with lower LAI (and therefore generally higher $I_L$), the leaf sampling method described will produce more sunlit leaves, while in higher LAI canopies (lower $I_L$), the sampled leaves are more likely to include leaves that develop in the shade. It is therefore not clear to what extent the results shown indicate variation in Narea with changes in top-of-canopy environment, vs. within-canopy changes in Narea driven by the well-studied vertical gradient in LMA. This issue is mentioned in the current revision in the methods section where estimation of $I_L$ is discussed, but

needs to be highlighted and explored as well in the results and discussion.

*The reviewer repeats here the assumption, which we dispute, that within-canopy changes in $N_{area}$ are "driven" by the vertical gradient of LMA. But the reviewer's main point, as we interpret this comment, is that our analysis did not distinguish responses of $N_{area}$ to top-of-canopy conditions from responses of $N_{area}$ to vertical gradients. That is so, because we were lacking data on the canopy position of each species. But we did analyse the response of (log-transformed) $N_{area}$ to (log-transformed) top-of-canopy PAR, and we found it was strong; with a slope > 1, as we would expect, given that the lower-PAR environments along this transect are also those with denser vegetation and therefore, as the reviewer observed, the sampling is likely to yield a greater number of leaves that developed in the shade. When we transformed top-of-canopy PAR to $I_L$ however the slope became close to 1 as theoretically predicted. We now mention this aspect in Results, section 3.1(page 9, lines 8) and Discussion, section 4.1(page 11, lines 16) as follow.*

*Revised in results, section 3.1 (page 9, lines 8),*

(A slope significantly greater than unity was found for ln $N_{area}$ versus ln $I_0$, i.e. top-of-canopy PAR, as expected as this measure underestimates the change in mean canopy PAR along the gradient from sparse, high-PAR to dense, lower-PAR communities.)

*Revised in discussion, section 4.1(page 11, lines 16)*

The relationship to site mean irradiance had a slope as predicted by the co-ordination hypothesis (i.e. close to 1) but a strong relationship, with a steeper slope as expected, was found when top-of-canopy irradiance was used instead of the canopy mean – indicating that both spatial variations and within-canopy shading were contributing to the relationship with site mean irradiance.

Minor points:

(p.2, l.6-7) Meaning could be clarified by rewording as "...should decrease with increases in ci:ca...", and "...declines with increases in both."

*We have implemented this improvement.*

(p.6, l.20) Were K and gamma* evaluated using site-level measurements?

*No. In any case we would not expect systematic site-to-site differences in the value of these quantities apart from the temperature dependencies of both quantities, which we consider explicitly. Instead, we used standard values. This is now stated, and we cite the appropriate reference as follows (in page 7, lines 14).*

[revised manuscript text omitted]